# A Study on Dimensionality Reduction and Parameters for Hyperspectral Imagery Based on Manifold Learning

**DOI:** 10.3390/s24072089

**Published:** 2024-03-25

**Authors:** Wenhui Song, Xin Zhang, Guozhu Yang, Yijin Chen, Lianchao Wang, Hanghang Xu

**Affiliations:** 1College of Geoscience and Surveying Engineering, China University of Mining and Technology (Beijing), Beijing 100083, China; bqt1800205061@student.cumtb.edu.cn (W.S.); bqt1900205063@student.cumtb.edu.cn (L.W.); bqt2000205068@student.cumtb.edu.cn (H.X.); 2Institute of Remote Sensing and Digital Earth, Chinese Academy of Sciences, Beijing 100101, China; zhangxin000181@aircas.ac.cn; 3State Grid General Aviation Co., Ltd., Beijing 102209, China; gzyang3912@163.com

**Keywords:** hyperspectral imagery, manifold learning, dimensionality reduction, feature extraction, optimal neighborhood, intrinsic dimensionality

## Abstract

With the rapid advancement of remote-sensing technology, the spectral information obtained from hyperspectral remote-sensing imagery has become increasingly rich, facilitating detailed spectral analysis of Earth’s surface objects. However, the abundance of spectral information presents certain challenges for data processing, such as the “curse of dimensionality” leading to the “Hughes phenomenon”, “strong correlation” due to high resolution, and “nonlinear characteristics” caused by varying surface reflectances. Consequently, dimensionality reduction of hyperspectral data emerges as a critical task. This paper begins by elucidating the principles and processes of hyperspectral image dimensionality reduction based on manifold theory and learning methods, in light of the nonlinear structures and features present in hyperspectral remote-sensing data, and formulates a dimensionality reduction process based on manifold learning. Subsequently, this study explores the capabilities of feature extraction and low-dimensional embedding for hyperspectral imagery using manifold learning approaches, including principal components analysis (PCA), multidimensional scaling (MDS), and linear discriminant analysis (LDA) for linear methods; and isometric mapping (Isomap), locally linear embedding (LLE), Laplacian eigenmaps (LE), Hessian locally linear embedding (HLLE), local tangent space alignment (LTSA), and maximum variance unfolding (MVU) for nonlinear methods, based on the Indian Pines hyperspectral dataset and Pavia University dataset. Furthermore, the paper investigates the optimal neighborhood computation time and overall algorithm runtime for feature extraction in hyperspectral imagery, varying by the choice of neighborhood k and intrinsic dimensionality d values across different manifold learning methods. Based on the outcomes of feature extraction, the study examines the classification experiments of various manifold learning methods, comparing and analyzing the variations in classification accuracy and Kappa coefficient with different selections of neighborhood k and intrinsic dimensionality d values. Building on this, the impact of selecting different bandwidths t for the Gaussian kernel in the LE method and different Lagrange multipliers λ for the MVU method on classification accuracy, given varying choices of neighborhood k and intrinsic dimensionality d, is explored. Through these experiments, the paper investigates the capability and effectiveness of different manifold learning methods in feature extraction and dimensionality reduction within hyperspectral imagery, as influenced by the selection of neighborhood k and intrinsic dimensionality d values, identifying the optimal neighborhood k and intrinsic dimensionality d value for each method. A comparison of classification accuracies reveals that the LTSA method yields superior classification results compared to other manifold learning approaches. The study demonstrates the advantages of manifold learning methods in processing hyperspectral image data, providing an experimental reference for subsequent research on hyperspectral image dimensionality reduction using manifold learning methods.

## 1. Introduction

### 1.1. Characteristics and Challenges of Hyperspectral Remote-Sensing Images

Remote sensing employs modern vehicles and sensors to acquire the electromagnetic characteristics of target objects from a distance, analyzing the shape, location, properties, and status changes of the targets through the transmission, storage, correction, and interpretation of information [1,2,3]. Among these, hyperspectral remote-sensing technology, characterized by high spectral resolution, high feature dimensionality, precise quantitative analysis, rich spectral information, and integrated imaging and mapping, is one of the emerging directions in remote-sensing science [4]. Hyperspectral remote-sensing technology significantly enriches the informational content of Earth observation. The reflectance of surface materials at different bands represents their spectral characteristics. Different land covers possess unique spectral features. Through steps such as spectral feature extraction, data analysis, and application, accurate identification and monitoring of surface features are achieved, enhancing the breadth and depth of applications in the surveying and mapping field. Unlike multispectral remote sensing, hyperspectral remote sensing can capture information across hundreds of continuous spectral bands on the Earth’s surface, providing rich spectral information to enhance the discriminative capability for different materials [5].

The core advantage of hyperspectral remote sensing lies in its ability to reflect subtle differences in spectral characteristics, but the vast number of bands presents significant challenges for data processing. The presence of many bands (or channels) within each pixel poses challenges for traditional land cover classification methods, which makes it difficult to apply learning algorithms directly [6]. Therefore, the “big data” characteristic leads to the “curse of dimensionality” issue, prone to the “Hughes” phenomenon; the strong correlation among bands causes high “information redundancy”; and the nonlinear structural features increase computational complexity.

(1)Curse of Dimensionality

The concept of the curse of dimensionality was first introduced by Richard E. Bellman in 1961, used to describe the problems faced in data analysis and organization within high-dimensional spaces. In hyperspectral imagery, this leads to data analysis and classification tasks becoming extremely challenging, with the algorithm’s classification performance exhibiting a “first increase then decrease” phenomenon as the feature dimensionality increases, a phenomenon known as the “curse of dimensionality” [7,8,9,10], also referred to as the “Hughes” phenomenon [11]. Although the impact of the “Hughes phenomenon” on classification accuracy gradually diminishes with the continued increase in the number of training samples, it faces the challenges of acquiring a large number of training samples and the cost of high computational complexity [12,13].

(2)Strong Inter-band Correlation

Similarity in spectral characteristics results in high correlation among adjacent bands due to their close wavelengths, reflecting very similar terrestrial spectral properties. This high correlation is further compounded by the continuous spectral response of materials, making the spectral information captured by neighboring bands in hyperspectral imagery often very similar, leading to high correlation. To enhance spectral coverage and sensitivity, the spectral response ranges of various bands might overlap, causing the information captured by adjacent bands to partially coincide, thus increasing inter-band correlation. A pixel covering a surface area on Earth might contain multiple materials, resulting in similar reflection properties among different bands. The acquisition of hyperspectral data is influenced by atmospheric conditions, such as the absorption and scattering effects of water vapor, carbon dioxide, and other components in the atmosphere on specific wavelengths, creating correlations among these bands. Factors like solar elevation and cloud cover also have a uniform effect on all bands of the hyperspectral imagery, further enhancing inter-band correlation. The strong correlation among bands leads to low algorithmic efficiency [14,15,16].

(3)Nonlinear Data Structures

The data within hyperspectral imagery are not distributed in a linear Euclidean space but rather in some form of nonlinear feature space. This includes nonlinear relationships in spectral reflectance, spectral mixing and interaction effects, nonlinear characteristics of land cover boundaries, nonlinear spatial correlations, and noise along with nonlinear variations, among others. It is essential to consider these nonlinearities in hyperspectral data. The collection process of hyperspectral remote-sensing data exhibits these nonlinear issues due to various environmental factors, such as different atmospheric components, differences in electromagnetic wave reflection angles, and the state of imaging system firmware, leading to nonlinear ground scattering. These are represented by models such as the Bidirectional Reflectance Distribution Function (BRDF) [17,18], noticeable nonlinear changes in the wavelength of minimum reflectance [18], attenuation effects of water body changes within a pixel [19], and the heterogeneity of multiple scattering and subpixel components within a pixel [20,21], rendering the spectral data of hyperspectral imagery nonlinear [22].

In summary, the “curse of dimensionality” within hyperspectral imagery presents a challenge related to data analysis and processing, especially in the fields of remote sensing and image recognition. The foundational issue is how to perform feature extraction on hyperspectral imagery, with dimensionality reduction being a necessary processing step. This paper will explore the capability and effectiveness of feature extraction in hyperspectral imagery based on manifold learning.

Building on this foundation, this paper further delves into the issue of parameter selection in manifold learning for feature extraction in hyperspectral imagery: one aspect is the estimation of the intrinsic dimensionality d, where values that are either too high or too low are detrimental to uncovering the nature of the data and important data characteristics. The second issue pertains to constructing the optimal neighborhood: it is critical to control the extent of the neighborhood graph, as overly extensive neighborhoods can lead to “short-circuiting” phenomena, and neighborhoods that are too restrictive may result in “disconnection” issues. Therefore, selecting an appropriate neighborhood size k and intrinsic dimensionality d is essential to unearth the latent structures of hyperspectral images and facilitate applications such as land cover classification.

There has been some research on determining the dimensionality of low-dimensional embedding space in manifold learning [23,24,25,26,27,28,29,30,31], but the question of how to more accurately determine the dimensionality of the low-dimensional space for hyperspectral imagery remains a topic worthy of in-depth study. Furthermore, the effects of different manifold learning methods on hyperspectral image feature extraction and the impact of manifold learning parameters on the feature extraction outcomes remain largely unexplored, as does the pattern of how manifold learning method parameter choices affect neighborhood computation and overall algorithm runtime. Based on these issues, the primary objectives of this study are: 1. to investigate and compare the applicability of various manifold learning algorithms in the dimensionality reduction of hyperspectral images; 2. to explore the impact patterns of parameters such as neighborhood size k and intrinsic dimensionality d on the feature extraction results of hyperspectral images using Isomap, LLE, LE, HLLE, LTSA, and MVU algorithms, as well as the effects of Gaussian kernel function bandwidth t and local structure weight λ on the feature extraction outcomes of the LE and MVU algorithms, respectively; 3. to study the influence of parameter neighborhood size k and intrinsic dimensionality d on the neighborhood computation time and overall runtime of the Isomap, LLE, LE, HLLE, LTSA, and MVU algorithms. This research significantly contributes to the use of manifold learning for feature extraction from hyperspectral images, offering substantial guidance and reference value.

### 1.2. Hyperspectral Manifold Learning Dimensionality Reduction and Related Parameter Mathematical Expression

In mathematics, it is common to establish a homeomorphism between manifolds and Euclidean spaces, allowing the combination of local Euclidean coordinate charts to substitute for the manifold, using the coordinates of the local Euclidean charts to identify elements on the manifold. For a given high-dimensional observational dataset in practical applications, variations among data can be represented by a few influencing factors. Statistically, the correlations between these factors are geometrically manifested as being scattered on a low-dimensional smooth manifold, where the number of influencing factors corresponds to the dimensionality of the smooth manifold. Here, we first present several mathematical definitions as follows:

**Definition 1.** *Assuming the hyperspectral image data are*  X∈ℝn×p*, the goal of manifold learning methods is to find a mapping:*  f:X→Y*, where*  Y∈ℝn×dd≪p  *represents the data’s representation on a low-dimensional manifold. The objective is to minimize some distance measure between points in the original data space and the low-dimensional embedding space. Here, n represents the number of pixels, and p represents the number of bands.*

**Definition 2.** *Let*  
M  *be a Hausdorff space. If for any point*  
x∈M*, there exists a* 
U *neighborhood* 
x *in* 
M *that is homeomorphic to an open set in* 
$R^{d}$*, then* 
M *is called a* 
d*-dimensional manifold, denoted as* 
dim(M)=d.

**Definition 3.** *Given a dataset*  
$X=\left\{x_{1},x_{2},\cdots ,x_{n}\right\}$*, where each*  
$x_{i}$ *is a point in the original high-dimensional space, for each point*  
$x_{i}$ *in the dataset, it holds that:*
(1)$$N_{k}\left(x_{i}\right)=\left\{x_{j}:dist\left(x_{i},x_{j}\right)\leq dist\left(x_{i},x_{\left(k\right)}\right)\right\}$$*where neighborhood* $N_{k}\left(x_{i}\right)$ *refers to the set of* 
k *nearest points to* 
$x_{i}$*,* 
$dist\left(x_{i},x_{j}\right)$ *is the distance between points* 
$x_{i}$ *and* 
$x_{j}$*, and* 
$x_{\left(k\right)}$ *is the distance to the* 
k*th nearest neighbor from* 
$x_{i}$.

Thus, estimating the dimensionality of the smooth manifold embedding space and selecting neighborhoods are key issues in studying meaningful low-dimensional manifold structures in high-dimensional data. Based on these two issues, this article primarily investigates the feature extraction capabilities and effects of different manifold learning methods on hyperspectral images. It explores the patterns of choosing different neighborhoods k and intrinsic dimensions d and their roles in hyperspectral image feature extraction. This provides a reference for future researchers using manifold learning for optimal neighborhood construction and determining potential dimensions in hyperspectral image feature extraction.

### 1.3. Manifold Expression in Hyperspectral Imagery Dimensionality Reduction

This section will formalize and express the feature extraction process of hyperspectral imagery data using manifold learning through the theories of manifold and differential geometry. The objective of manifold learning is to maintain the manifold structure of the data as much as possible while reducing its dimensionality. This method can discover the local linear relationships among samples in high-dimensional space and map these relationships to a low-dimensional space. Thus, the principle of manifold learning feature extraction for hyperspectral imagery is to map the data from a high-dimensional space to a low-dimensional space, obtaining a compact low-dimensional representation of the original dataset. Hyperspectral imagery contains a large number of bands, each corresponding to spectral information at different wavelengths. These bands together constitute a multi-dimensional data space. Due to physical and geographical constraints, even though the data itself are high-dimensional, there exists dependency among the pixels, implying that the data may be confined within a low-dimensional manifold. Therefore, hyperspectral imagery data can be considered a high-dimensional representation of a low-dimensional manifold space, with each pixel representing a point within the manifold space. Similarly, similar land cover types (such as water bodies, vegetation, soil, etc.) will form clusters in the spectral feature space, which can be viewed as neighborhoods of points on the manifold.

The manifold learning dimensionality reduction process in hyperspectral imagery reduction involves the following:

First, a formal definition of a coordinate chart $\left(U,\varphi \right)$ is given for the hyperspectral imagery spectral feature dataset.

Then, based on the definitions of differential manifold tangent vectors and tangent spaces, a formal definition is provided for the local neighborhood corresponding to tangent spaces of the hyperspectral imagery spectral feature manifold space dataset.

Finally, based on the definition of the differential manifold’s tangent bundle, we present the derivation of global low-dimensional manifold coordinates from local tangent coordinates in the hyperspectral image manifold space, achieving the extraction of spectral features and dimensionality reduction in hyperspectral images.

**Definition 4** (Local Coordinate Neighborhood of a Hyperspectral Data Sample Point)**.** *If a sample point* 
$x_{i}$ *in the hyperspectral dataset is considered as a point within coordinate chart* 
U*, then* 
U *is the open set of the neighborhood where sample point* 
$x_{i}$ *is located, with the set* 
U *consisting of elements* 
$\left\{x_{1},x_{2},x_{3}\cdots x_{i}\right\}$*, where* 
$x_{i}$ *represents the data sample points in the local neighborhood, then the set* 
U *is referred to as the local coordinate neighborhood of the hyperspectral data sample point.*

From the manifold learning dimensionality reduction process, it is understood that in practical research, all pixels within a certain range of a pixel in hyperspectral remote-sensing data are considered as the optimal neighborhood of that point in the manifold space; by calculating the minimum reconstruction error function, the corresponding local tangent space coordinates are obtained. In the manifold space of the original hyperspectral remote-sensing data, multiple sets of local tangent space coordinates representing each pixel point’s coordinate neighborhood are calculated; finally, by calculating the global optimal reconstruction error, the overlapping local tangent space coordinate sets are arranged to obtain the global manifold coordinates of the dimensionality-reduced hyperspectral remote-sensing imagery, achieving dimensionality reduction of hyperspectral remote-sensing data. The manifold learning expression is as follows:

**Definition 5.** *Let the manifold space in which the hyperspectral imagery dataset resides be denoted as* 
MHSI*, and let the dataset of sample points within a certain neighborhood centered on a particular pixel be an open covering* 
UHSI *of* 
MHSI*, with a corresponding family of continuous mappings* 
φHSI:UHSI→ℝd.

wherein UHSI represents the local coordinate neighborhood, indicating the optimal neighborhood of a certain pixel in hyperspectral remote-sensing imagery; φHSI denotes the local coordinate mapping, representing the Euclidean linear expression of the local nonlinear structure in hyperspectral remote-sensing imagery, achieving dimensionality reduction of hyperspectral imagery data; UHSI is the local coordinate covering of MHSI, indicating the calculated optimal neighborhood collection of hyperspectral remote-sensing imagery; UHSI,φHSI is a local coordinate system, representing the local coordinates of hyperspectral remote-sensing imagery obtained through the calculation of the optimal neighborhood; ℝd calculates the local tangent space obtained from the optimal neighborhood, representing the sub-feature space after dimensionality reduction of hyperspectral remote-sensing imagery (as shown in Figure 1).

## 2. Materials and Methods

### 2.1. Study Area

This paper selects the Indian Pines and Pavia University datasets as the experimental data foundation (as shown in Figure 2). The Indian Pines landscape contains 16 land cover categories, with a total of 10,249 pixels containing land cover. However, due to the non-reflectance of water in bands 104–108, 150–163, and 220, we typically use the remaining 200 bands, excluding these 20 bands, for research purposes. The Pavia University dataset encompasses 9 land cover classes with a total of 42,776 pixels containing land cover. Twelve bands are excluded due to noise interference, leaving 103 spectral bands for research. The dataset can be accessed through the following URL: https://ehu.eus/ccwintco/index.php/Hyperspectral_Remote_Sensing_Scenes (accessed on 9 January 2024).

This article evaluates the inter-band relationships within hyperspectral data based on correlation coefficients, calculated between the bands of hyperspectral imagery. For the Indian Pines dataset, the three bands with the lowest correlation, specifically bands 1, 88, and 188, exhibit a correlation coefficient of 0.0069. In the case of the Pavia University dataset, the three least correlated bands are 1, 62, and 103, with a correlation coefficient of 0.2243. These three bands are utilized to represent the red (R), green (G), and blue (B) channels, respectively, forming an RGB pseudocolor image (as shown in Figure 3).

The dataset consists of a total of 207,400 pixels, of which only 42,776 pixels are land cover pixels, whereas the remaining 164,624 pixels are background pixels. In practical classification tasks, these background pixels need to be excluded. The distribution of pixels across different land cover classes is presented in Table 2.

### 2.2. Data Description

In this study, we have selected 30% of each of the seven land cover types from the Indian Pines dataset and 10% of each of the eight land cover types from the Pavia University dataset (as shown in Table 3) for investigating manifold learning feature extraction and land cover classification experiments.

In this study, the division of the dataset for the land cover classification experiments is as follows: 80% for the training set and 20% for the validation set (as shown in Table 4). The classifier chosen for this task is Random Forest (RF), selected due to its excellent performance in handling high-dimensional data and its proven effectiveness in existing related research.

### 2.3. Methods

#### 2.3.1. Linear Manifold Learning Methods

(1)Principal Components Analysis (PCA)

PCA is a commonly used data dimensionality reduction technique [32,33]. It employs a linear transformation to map high-dimensional data to a lower-dimensional space, reducing the dimensionality of the data while retaining as much information as possible. The algorithm steps are shown in Algorithm 1 as follows:
**Algorithm 1** Principal Component Analysis (PCA)**Input:** A dataset **X**, intrinsic dimensionality d**Output:** Return low-dimensional coordinate matrix Y=y1,y2,⋯ydStep 1. Standardize the original dataset: Xstd=X−μδStep 2. Calculate the covariance matrix: Cov(Xstd)=1n−1XstdTXstdStep 3. Compute the eigenvalues and eigenvectors of the covariance matrix: Cov(Xstd)v=λvStep 4. Order the eigenvalues, select the principal components, construct the projection matrix, and transform into the new space: Xpca=XstdW


(2)Multidimensional Scaling (MDS)

MDS is a nonlinear technique used to embed high-dimensional data into a low-dimensional space. The basic idea is to project points from a high-dimensional coordinate system into a low-dimensional space, maintaining the similarity between points as much as possible. For this reason, in the low-dimensional space, the pairwise distances between points are very close to their actual distances. The algorithm steps are shown in Algorithm 2 as follows:
**Algorithm 2** Multidimensional Scaling (MDS)**Input:** A dataset **X**, intrinsic dimensionality d**Output:** Return low-dimensional coordinate matrix Y=y1,y2,⋯ydStep 1. Calculate a distance matrix D based on the original high-dimensional dataStep 2. Perform double centering on the distance matrix: B=−12JD2JStep 3. Conduct eigenvalue decomposition on the double-centered matrix:  Bvi=λiviStep 4. Select the principal components and compute the configuration:  X=VkΛk1/2

(3)Linear Discriminant Analysis (LDA)

LDA is a commonly used dimensionality reduction technique that considers the impact of categories during the reduction process, ensuring that samples from different classes have the maximum separation after dimensionality reduction. LDA aims to find an optimal linear projection where the variance within classes is minimized while the variance between classes is maximized. The algorithm steps are shown in Algorithm 3 as follows:
**Algorithm 3** Linear Discriminant Analysis (LDA)**Input:** A dataset **X**, intrinsic dimensionality d,data labels label**Output:** Return low-dimensional coordinate matrix Y=y1,y2,⋯ydStep 1. Calculate the within-class mean and the overall mean: μi=1ni∑x∈class ix and μ=1n∑i=1nxiStep 2. Compute the within-class scatter matrix and the between-class scatter matrix: SW=∑i=1kSi and SB=∑i=1kni(μi−μ)(μi−μ)TStep 3. Calculate the projection vector: SBv=λSWvStep 4. Select the eigenvector corresponding to the largest eigenvalue and transform into the new space: X′=XW


#### 2.3.2. Nonlinear Manifold Learning Methods

(1)Isometric Mapping (Isomap)

The core algorithm employed by the Isomap algorithm is consistent with MDS, with the distinction lying in the calculation of the distance matrix in the original space. As a significant improvement over the traditional MDS algorithm, Tenenbaum et al. [34] introduced the concept of “geodesic distance” in the Isomap, achieving the discovery of the low-dimensional manifold structure embedded in high-dimensional space by maintaining the geodesic distance unchanged between every two points in the high-dimensional dataset. The algorithm implementation steps are shown in Algorithm 4 as follows:
**Algorithm 4** Isometric Mapping (Isomap)**Input:** A dataset **X**, intrinsic dimensionality d, the neighborhood k**Output:** Return low-dimensional coordinate matrix Y=y1,y2,⋯ydStep 1. Calculate the double-centered distance matrix: B=−12JD2JStep 2. Perform eigenvalue decomposition on the double-centered matrix **B**, obtaining eigenvalues and their corresponding eigenvectorsStep 3. Select the eigenvectors corresponding to the largest k eigenvalues, which form the basis of the low-dimensional space Step 4. Use the selected eigenvectors and the square roots of their corresponding eigenvalues to compute the coordinates of the data points in the low-dimensional space

(2)Locally Linear Embedding (LLE)

LLE, proposed by Sam Roweis and others [35] from University College London, is based on the core idea that the low-dimensional manifold to be solved is locally linear, with each data point being representable as a linear combination of its neighbors. The process of manifold dimensionality reduction involves reconstructing the original data points by keeping the linear coefficients within each neighborhood constant, thereby minimizing the reconstruction error. The algorithm implementation steps are shown in Algorithm 5 as follows:
**Algorithm 5** Locally linear embedding (LLE)**Input:** A dataset **X**, intrinsic dimensionality d, the neighborhood k**Output:** Return low-dimensional coordinates matrix Y=y1,y2,⋯ydStep 1. Given dataset X=x1,x2,⋯,xN, for each point xi, find its k nearest neighbors xi1,xi2,⋯,xikStep 2. Calculate weights Wij based on the linear relationship ∑j=1kWijxi−xij=0Step 3. Find the low-dimensional representation Y=y1,y2,⋯yd by minimizing ΦY=∑i=1Nyi−∑j=1NWijyj2

(3)Laplacian Eigenmaps (LE)

LE, proposed by Belkin et al. [36], is based on the fundamental idea of preserving the local neighborhood relationships of data. It operates on the principle of eigenvalue decomposition of the graph Laplacian matrix, aiming to maintain the local proximity relations. This is achieved by constructing a graph representing the dataset, where the edge weights reflect the local distances between nodes (data points). The goal is to preserve the graph’s local adjacency relationships while re-drawing this graph in a lower-dimensional space from its high-dimensional origin. The algorithm implementation steps are shown in Algorithm 6 as follows:
**Algorithm 6** Laplacian Eigenmaps (LE)**Input:** A dataset **X**, intrinsic dimensionality d, the neighborhood k, the Gaussian kernel function bandwidth t**Output:** Return low-dimensional coordinate matrix Y=y1,y2,⋯ydStep 1. Based on the given dataset, construct an adjacency graph GStep 2. Compute the Laplacian matrix: L=D−WStep 3. Solve for the eigenvalues: Lv=λDvStep 4. Select the eigenvectors corresponding to the smallest non-zero eigenvalues of L as the coordinates of the data points in the low-dimensional space

(4)Hessian Locally Linear Embedding (HLLE)

Professor Donoho and colleagues from the Department of Statistics at Stanford University introduced HLLE [37]. The method involves finding neighboring points for each data point, then estimating the Hessian matrix within the neighborhood of each point. These local Hessian matrices are combined to form a global Hessian matrix. The essence of HLLE lies in preserving the local curvature of the data manifold, thereby better maintaining the data structure in a lower-dimensional space. By minimizing the second-order derivatives of the global Hessian matrix, HLLE finds a low-dimensional representation of the data. The algorithm implementation steps are shown in Algorithm 7 as follows:
**Algorithm 7** Hessian locally linear embedding (HLLE)**Input:** A dataset **X**, intrinsic dimensionality d, the neighborhood k**Output:** Return low-dimensional coordinate matrix Y=y1,y2,⋯ydStep 1. First, for each point in the dataset, find its K nearest neighbors and construct a K nearest neighbor graphStep 2. For each data point and its K nearest neighbors, estimate the Hessian matrix to reflect the curvature of the local geometric structureStep 3. By combining the local Hessian estimates of each point, construct a global Hessian matrix HStep 4. The space embedding involves eigenvalue decomposition and selection of the eigenvectors corresponding to the smallest non-zero eigenvalues: minYDTY=I trace(YHTY)


(5)Local Tangent Space Alignment (LTSA)

LTSA was proposed by Zhang et al. in 2004 [38]. The fundamental principle of LTSA is the assumption that data locally reside within the tangent space of a low-dimensional manifold. It estimates the local tangent spaces around each point and seeks a global low-dimensional embedding to optimally align these local tangent spaces. The algorithm aims to minimize the global reconstruction error of local tangent spaces, which is achieved by optimizing the following objective function: min ∑i=1N||Xi−∑j∈K(i)ωijXj||2. The steps for implementing the Algorithm 8 are:
**Algorithm 8** Local Tangent Space Alignment (LTSA)**Input:** A dataset **X**, intrinsic dimensionality d, the neighborhood k**Output:** Return low-dimensional coordinate matrix Y=y1,y2,⋯ydStep 1. For each point Xi in the dataset, find its K nearest neighbors and construct a K nearest neighbor graph Step 2. For each point and its K nearest neighbors, calculate the basis of the local tangent space through PCA of the local neighborhood Step 3. By rotating and translating each local tangent space, find a global reference frame that aligns all local tangent spaces within this frame as closely as possible Step 4. On the basis of the aligned local tangent spaces, reconstruct the global low-dimensional coordinates to preserve the geometric structure of local neighborhoods

(6)Maximum Variance Unfolding (MVU)

MVU, also known as Semi-definite Embedding (SDE), was proposed by Weinberger and Saul [39,40]. The fundamental idea is that the variance on the points is maximized when the manifold is correctly unfolded, aiming to keep the data points as far apart from each other as possible in a low-dimensional representation, thereby maximizing the variance of the data. The algorithm implementation steps are shown in Algorithm 9 as follows:
**Algorithm 9** Maximum Variance Unfolding (MVU)**Input:** A dataset **X**, intrinsic dimensionality d, the neighborhood k, the lambda λ**Output:** Return low-dimensional coordinate matrix Y=y1,y2,⋯ydStep 1. For each point Xi in the dataset, identify its nearest neighbors and construct a K nearest neighbor graphStep 2. Calculate and store the Euclidean distances between the data point and its K nearest neighborsStep 3. Maximize the variance in the low-dimensional space through semidefinite programming (SDP): maxY Tr(YTY)
Step 4. Use a semidefinite programming (SDP) solver to solve this optimization problem and extract the low-dimensional embedding from the solution of SDP, selecting the eigenvectors corresponding to the largest few eigenvalues as the coordinates in the low-dimensional space

## 3. Results

### 3.1. Visualization of Low-Dimensional Embedding of Hyperspectral Images

To validate the performance of manifold learning algorithms in feature extraction within hyperspectral imagery, the dataset from Table 3 was visualized by selecting 30% of the total sample points from each of the seven land cover types and projecting them into three-dimensional space (first characteristic, second characteristic, and third characteristic). The scatter plot after dimensionality reduction is shown in Figure 4. The LDA algorithm demonstrates good inter-class and intra-class separability due to its use of class labels to guide dimensionality reduction, hence exhibiting better performance in classification and recognition tasks. However, it fails to reveal the intrinsic nonlinearity of hyperspectral features, merely showcasing the linear relationships of data separability, leaving gaps in the extraction and understanding of hyperspectral spectral features. The PCA and MDS algorithms have poorer inter-class separability, with clear separation between corn and forest in the boundary areas, but other categories almost entirely overlap, due to these algorithms not adequately considering local features. The Isomap algorithm shows good separability for some categories but poor separability for others, and through visualization, it is clear that the Isomap results reflect the global properties of hyperspectral features well, determined by the algorithm’s acquisition of global coordinates, and the use of geodesic distance to calculate the optimal neighborhood also reflects the fact that locally close data points can adhere better. The LLE algorithm exhibits good intra-class and inter-class separability, achieving better classification to a certain extent. The LE algorithm shows poor separability, even featuring a loss of characteristics, related to the setting of the σ value when constructing distance weights using the Gaussian kernel function. The LTSA and HLLE algorithms demonstrate certain inter-class separability, with poorer intra-class separability, but clearly show the nonlinear relationships between high spectral features, determined by the algorithm’s maintenance of local features and local curvature. MVU results are similar to those from PCA and MDS, and although MVU is a nonlinear manifold learning method, it employs the PCA method in constructing local coordinates before generalizing to the global; hence, MVU can essentially be considered a locally linear, globally nonlinear manifold learning method, which is why its results are similar to those of PCA and MDS. Therefore, it can be concluded that nonlinear manifold learning methods surpass linear manifold learning methods in uncovering the intrinsic nonlinear relationships of hyperspectral imagery spectral features, with LTSA, LLE, and HLLE specifically demonstrating the nonlinear relationships of hyperspectral imagery spectral features more effectively. The subsequent sections will delve deeper and more comprehensively into the study of different manifold learning methods for feature extraction and dimensionality reduction in hyperspectral imagery.

### 3.2. Experimental Results and Comparative Analysis Based on the Indian Pines Dataset

Based on the selected Indian Pines dataset, this study investigates the feature extraction of hyperspectral imagery using linear and nonlinear manifold learning methods. The spectral features extracted after dimensionality reduction are used for land cover classification experiments and analysis. Through overall accuracy, Kappa coefficient, neighborhood computation time, and overall algorithm runtime, the study analyzes and compares the dimensionality reduction capabilities and feature extraction effectiveness of different manifold learning methods when selecting various neighborhoods and intrinsic dimensions. Additionally, the study examines the quality of hyperspectral imagery feature extraction and the effects of dimensionality reduction for the LE and MVU algorithms when different values of t and λ are chosen.

#### 3.2.1. Linear Manifold Learning Dimensionality Reduction in Hyperspectral Imagery

As the intrinsic dimension d increases, the classification accuracy and Kappa coefficient for PCA, MDS, and LDA change. From Figure 5, it can be observed that with the continuous increase in the reduced dimension d, the classification accuracy and Kappa coefficient of the three methods also increase. However, when d = 10, the classification accuracy and Kappa coefficient remain essentially unchanged, indicating that the potential dimensions of spectral features mined using the PCA, MDS, and LDA linear manifold learning methods for this dataset are basically consistent. Through classification accuracy, it can be intuitively seen that the accuracy from low to high is sequentially PCA, MDS, LDA, which suggests that using LDA for hyperspectral imagery feature extraction can achieve better dimensionality reduction effects. On the other hand, the algorithm execution time consumption indicates that the MDS algorithm consumes the most time, whereas LDA consumes the least. Therefore, combining classification accuracy and time consumption, it is clear that the LDA algorithm outperforms the other two methods in hyperspectral imagery feature extraction. However, linear manifold learning algorithms cannot intervene in the processing through parameterization or other means, requiring users to have some prior knowledge of the observation object and grasp some characteristics of the data; the importance of each dimension is the same, unable to distinguish the importance of different dimensions, and this is not suitable for nonlinear dimensionality reduction problems.

Compared to unsupervised dimensionality reduction methods like PCA and MDS, LDA uses labeled sample information to train and obtain the optimal projection direction, making the projected samples of the same class as compact as possible. Therefore, it exhibits better performance in classification and recognition tasks. However, like PCA and other linear dimensionality reduction methods, it also seeks the projection direction through finding the linear projection of the original data; thus, it is equally ineffective for data with nonlinear structures.

#### 3.2.2. Nonlinear Manifold Learning Dimensionality Reduction in Hyperspectral Imagery

Traditionally, dimensionality reduction has been performed using linear techniques such as PCA. In recent years, a plethora of nonlinear techniques for dimensionality reduction has been proposed [41,42,43]. This section will investigate and evaluate the performance of the Isomap, LLE, LTSA, HLLE, LTSA, and MVU nonlinear manifold learning methods in hyperspectral imagery feature extraction and dimensionality reduction through the selection of neighborhood k, intrinsic dimension d, Gaussian kernel function parameter t, and Lagrange multiplier λ, using land cover classification accuracy (overall accuracy, OA) and Kappa coefficient values and their patterns of variation.

Comparison of classification accuracy in nonlinear manifold learning

(1) Comparison of classification accuracy across manifold learning methods with different selections of k and d.

As shown in Figure 6a–d, selecting different neighborhoods k = 10, 30, 50, 80, 120, and 150, and with the continuous increase in intrinsic dimension d, the patterns of change in land cover classification accuracy and Kappa coefficient after dimensionality reduction of hyperspectral imagery using the Isomap, LLE, HLLE, and LTSA methods are observed. With the continuous increase in k and d, the land cover classification accuracy after feature extraction using nonlinear manifold learning methods keeps increasing. For the Isomap, LLE, and HLLE methods, when d = 20, the classification accuracy and Kappa coefficient change slowly, and different k values have a minor impact on classification accuracy with the continued increase in d, indicating the optimal intrinsic dimension d = 20 for these three methods across different k values; for the LTSA method, when d = 30, the classification accuracy and Kappa coefficient change slowly, and different k values have a minor impact on classification accuracy with the continued increase in d, indicating that the optimal intrinsic dimension for the LTSA method is d = 30 across different k values. It can also be seen that the land cover classification accuracy after feature extraction using LTSA is superior to Isomap, LLE, and HLLE. In summary, the Isomap, LLE, HLLE, and LTSA methods can all achieve feature extraction from hyperspectral imagery, but in terms of classification accuracy, the LTSA method outperforms the other three nonlinear manifold learning methods.

In the LE algorithm, when k is set to different values, the classification accuracy and Kappa coefficient change with the continuous increase in intrinsic dimension d for t = 1, 50, 100, 150, 200, and 300 (as shown in Figure 7). Regardless of the value of k, when t = 1, the land cover classification accuracy after feature extraction by the LE method remains almost unchanged; with a certain k value, as t continuously increases, so does the classification accuracy; with certain values of k and t, as d increases, the classification accuracy also increases, and after d = 10, the classification accuracy maintains an equilibrium; with a fixed d, as t increases, classification accuracy continues to increase, but when t = 120, the classification accuracy no longer continues to increase; hence, it can be seen that when d = 10, with the increase in d and k, the classification accuracy almost does not change, but it continuously increases with the increase in t.

This is due to the fact that in the LE algorithm, the parameter t is used to control the scaling of the distances in the Gaussian kernel, which affects the computation of the inter-sample weights. When the value of t is small, neighbors very close to the centroid have a significant effect on the centroid, thus emphasizing the local structure of the data. When the t value is large, the Gaussian kernel decays more slowly, thus emphasizing the global structure of the data. Therefore, the size of the t value needs to be set flexibly according to the research needs.

In the MVU algorithm, a key role is played by λ, which represents the importance of the distance invariance constraint between each nearest neighbor pair (i.e., pairs of points that are close together in the high-dimensional space). If λ is too large, it may lead to a low-dimensional representation that is too dependent on the local neighborhood structure of the original data, and λ is too small to emphasize the global neighborhood structure. Thus, inappropriate values may cause the algorithm to have difficulty converging or converging to a local optimum.

In the MVU algorithm, when k is set to different values, the classification accuracy and Kappa coefficient change with the continuous increase in intrinsic dimension d for λ = 0.5, 1, 5, and 10 (as shown in Figure 8). With a certain λ value, as k continuously increases, so does the classification accuracy, which then changes slowly when k = 80; with certain k and λ values, as d increases, the classification accuracy also increases, and after d = 10, the classification accuracy maintains equilibrium; with a fixed d, as λ increases, the classification accuracy also slowly increases, but with negligible difference.

(2) Comprehensive analysis of classification accuracy across manifold learning methods with different selections of k and d.

As illustrated in Figure 9a–f, selecting different neighborhoods k = 10, 30, 50, 80, 120, and 150, and with the continuous increase in intrinsic dimension d, the study compares and contrasts the changing patterns of land cover classification accuracy and Kappa coefficient using the Isomap, LLE, LTSA, HLLE, LTSA, and MVU nonlinear manifold learning methods in hyperspectral imagery. With a certain k value, as the intrinsic dimension d continuously increases, the land cover classification accuracy of the Isomap, LLE, LTSA, HLLE, LTSA, and MVU nonlinear manifold learning methods also continuously increases and eventually reaches a balance. However, it is observable that the intrinsic dimension d at which each nonlinear manifold learning method’s classification accuracy reaches its equilibrium value differs, indicating that the intrinsic dimensions of hyperspectral imagery features derived from different manifold learning methods are not consistent. Moreover, it is seen that the classification accuracy obtained by the LTSA method is slightly higher than that of the other five nonlinear manifold learning methods; as the k value continuously increases, the land cover classification accuracy and Kappa coefficient of each nonlinear manifold learning method also continuously increase, which also indicates the impact of neighborhood selection on the results of manifold learning feature extraction. Therefore, to achieve ideal feature extraction results with any nonlinear manifold learning method, it is necessary to select appropriate neighborhood k and intrinsic dimension d values.

From the confusion matrices of each algorithm, as shown in Figure 10, it is evident that the LTSA algorithm surpasses other manifold learning methods in terms of overall classification accuracy as well as the classification accuracy for each type of land cover. This indicates that the LTSA algorithm outperforms other manifold learning methods in both overall and individual land cover classification accuracies.

2.Comparison of neighborhood computation time in nonlinear manifold learning

As shown in Figure 11, selecting different neighborhoods k = 50, 120, and 150, and with the continuous increase in intrinsic dimension d, the study investigates and compares the neighborhood computation time for dimensionality reduction in hyperspectral imagery using the Isomap, LLE, LTSA, HLLE, LTSA, and MVU manifold learning methods. When a certain k value is chosen, the neighborhood computation time for Isomap, HLLE, and MVU increases with the increase in dimension d, indicating that dimension d affects their neighborhood computation. Conversely, for LLE, LE, and LTSA, the neighborhood computation time remains balanced with the increase in dimension d, suggesting that dimension d does not affect their neighborhood computation. With a certain d value, the neighborhood computation time for the Isomap, LLE, LTSA, HLLE, LTSA, and MVU manifold learning methods also increases with the increase in k, indicating that k influences computation. It can also be found that regardless of the continuous increase and changes in neighborhood k and intrinsic dimension d, the neighborhood computation time for LE remains almost unchanged, indicating that the LE method has the lowest time complexity among all nonlinear manifold learning methods. However, it can also be observed that the neighborhood computation time for HLLE increases the most with the continuous increase and changes in neighborhood k and intrinsic dimension d, indicating that it has the highest computational complexity among all manifold learning methods.

3.Comparison of algorithm runtime in nonlinear manifold learning

As illustrated in Figure 12, selecting different neighborhoods k = 50, 120, and 150, and with the continuous increase in intrinsic dimension d, the study investigates and compares the algorithm runtime for dimensionality reduction in hyperspectral imagery using the Isomap, LLE, LTSA, HLLE, LTSA, and MVU nonlinear manifold learning methods. When a certain k value is chosen, the runtime of the Isomap, HLLE, LTSA, and MVU methods continuously increases with the increase in intrinsic dimension d, and both the Isomap and MVU methods experience a significant increase in time consumption with the increase in d, whereas the LLE, LE, HLLE, and LTSA methods show a smaller increase; with a certain d value, as the neighborhood k continuously increases, the runtime of the Isomap and HLLE methods increases more rapidly, the LLE, LTSA, and MVU methods increase more slowly, and the LE method’s runtime remains almost unchanged. It can also be seen that regardless of different selections of k and d, the overall runtime of the HLLE method is significantly higher than that of the other five nonlinear manifold learning methods, indicating that HLLE has the highest algorithmic complexity.

4.Comparison of neighborhood and algorithm runtime in nonlinear manifold learning

As shown in Figure 13a–f, selecting different neighborhoods k = 10, 50, 80, 120, and 150, and with the continuous increase in intrinsic dimension d, the study investigates and compares the neighborhood and algorithm runtime for dimensionality reduction in hyperspectral imagery using the Isomap, LE, LLE, HLLE, LTSA, and MVU nonlinear manifold learning methods.

Figure 13a,b reveal that with a fixed d value, as k increases, both neighborhood computation and algorithm runtime for the Isomap and LE methods increase, and the runtime is consistently greater than the neighborhood computation time, indicating that neighborhood computation time has a relatively small impact on the overall algorithm runtime during dimensionality reduction; with a fixed k value, as dimension d continuously increases, both neighborhood computation and algorithm runtime increase slowly, suggesting that dimension d has a minor impact on algorithm time cost.

Figure 13c shows that with a fixed d value, as k increases, both neighborhood computation and algorithm runtime increase, and the difference between them is small, indicating that the LLE method’s neighborhood computation significantly impacts the overall algorithm runtime cost during dimensionality reduction; with a fixed k value, as dimension d continuously increases, both neighborhood computation and algorithm runtime increase slowly, suggesting that dimension d has a minor impact on algorithm time cost.

Figure 13d illustrates that with a fixed d value, as k increases, both neighborhood computation and algorithm runtime increase, and the difference between them is significant, indicating that the HLLE method’s neighborhood computation time is far less than the overall algorithm runtime cost; with a fixed k value, as dimension d continuously increases, both neighborhood computation and algorithm runtime increase slowly, suggesting that dimension d has a minor impact on algorithm time cost.

Figure 13e indicates that with a fixed d value, as k increases, both neighborhood computation and algorithm runtime increase, and the difference between them is stable, indicating that the LTSA method’s neighborhood computation time and overall algorithm runtime cost are comparable; with a fixed k value, as dimension d continuously increases, both neighborhood computation and algorithm runtime increase slowly, suggesting that dimension d has a minor impact on algorithm time cost.

Figure 13f demonstrates that with a fixed d value, as k increases, both neighborhood computation and algorithm runtime increase, and the difference between them continuously grows, indicating that the MVU method’s neighborhood computation time has an increasingly significant impact on the overall algorithm with the continuous increase in k; with a fixed k value, as dimension d continuously increases, both neighborhood computation and algorithm runtime increase slowly, and the difference between them is significant and tends to stabilize, suggesting that only when dimension d is large does it impact the algorithm runtime.

The aforementioned research content compares the performance of six nonlinear manifold learning methods, Isomap, LE, LLE, HLLE, LTSA, and MVU, in dimensionality reduction of hyperspectral imagery, which can be summarized as follows:

The neighborhood computation time for the Isomap, LE, and HLLE methods is less than the total algorithm runtime, indicating that the impact of neighborhood computation time on the overall algorithm runtime is relatively minor. In contrast, for the LLE algorithm, the neighborhood computation time occupies a significant proportion of the total algorithm runtime, thus having a considerable impact on the overall algorithm performance. The LTSA algorithm shows a stable difference between neighborhood computation time and total runtime, suggesting that the expenses for neighborhood computation and overall algorithm runtime are approximately equal. For the MVU algorithm, the discrepancy between neighborhood computation time and total runtime is pronounced, indicating an increasingly significant impact of neighborhood computation time on the total algorithm runtime. Comparing these nonlinear manifold learning methods reveals that both neighborhood computation time and algorithm runtime are significantly influenced by the size of the neighborhood, whereas the increase in intrinsic dimensionality has a relatively smaller effect on the time expenditure.

### 3.3. Experimental Results and Comparative Analysis Based on the Pavia University Dataset

This section of the experiment is conducted based on the selected Pavia University dataset, following the same research approach and process as that based on the Indian Pines dataset. This part focuses on studying feature extraction using various manifold learning methods applied to land cover classification experiments based on the Pavia University dataset. It also investigates the impact of neighborhood size k and intrinsic dimensionality d on the neighborhood computation time and overall runtime of each manifold learning algorithm. The purpose is to compare these results with the experimental results from the Indian Pines dataset to validate the universality and generalizability of the conclusions drawn in this paper.

#### 3.3.1. Linear Manifold Learning Dimensionality Reduction in Hyperspectral Imagery

As indicated in Figure 14, the land cover classification results based on the Pavia University dataset are consistent with the experimental outcomes from the Indian Pines dataset. However, a notable difference is that feature extraction using the MDS algorithm on the Pavia University dataset required the least amount of time, whereas the time expenditures for LDA and PCA were virtually indistinguishable. This suggests that the time cost of feature extraction using linear manifold learning methods varies when facing different datasets. Given that PCA, MDS, and LDA are all linear methods, and considering the inherent non-linear relationships within hyperspectral data, this also highlights the limitations of linear methods in processing non-linear data. Furthermore, it demonstrates that the non-linear relationships in hyperspectral data have a significant impact on linear methods.

#### 3.3.2. Nonlinear Manifold Learning Dimensionality Reduction in Hyperspectral Imagery

Comparison of classification accuracy in nonlinear manifold learning

(1) Comparison of classification accuracy across manifold learning methods with different selections of k and d.

For this part of the experiment, neighborhood sizes of k = 10, 30, 50, 80, and 100 were selected. As illustrated in Figure 15, the classification results after dimensionality reduction using the Isomap, LLE, HLLE, and LTSA methods follow the same pattern of variation with neighborhood size and intrinsic dimensionality as that observed with the results based on the Indian Pines dataset.

In this part of the experiment, the same value of t as that used in the Indian Pines dataset was selected. As shown in Figure 16, the land cover classification experiment after dimensionality reduction through the LE algorithm revealed that the LE algorithm’s sensitivity to the local weight t follows a consistent pattern with the Indian Pines dataset. This indicates that the influence of the t value on the LE algorithm during hyperspectral data dimensionality reduction is definite.

In this segment of the experiment, the same λ value as that used in the Indian Pines dataset was selected. As depicted in Figure 17, land cover classification experiments conducted with post-dimensionality reduction via the MVU algorithm demonstrated that the classification results after dimensionality reduction are inconsistently affected by the neighborhood size k and λ value across different hyperspectral datasets. Specifically, results from the Pavia University dataset indicated that when λ exceeds 5, an increase in neighborhood size k leads to a significant decrease in classification accuracy. This implies that the selection of neighborhood size k and λ significantly impacts the classification outcomes after dimensionality reduction when processing various datasets. Therefore, careful consideration of the values of neighborhood size k and λ is essential when employing the MVU algorithm for feature extraction from hyperspectral images for land cover classification experiments.

(2) Comprehensive analysis of classification accuracy across manifold learning methods with different selections of k and d.

As depicted in Figure 18, by selecting neighborhood sizes k = 10, 30, 50, 80, and 100 and varying the intrinsic dimensionality d, this study investigates and compares the changing patterns of land cover classification accuracy and the Kappa coefficient for hyperspectral images using the nonlinear manifold learning methods of Isomap, LLE, LTSA, HLLE, LTSA, and MVU. Experimental results indicate that, consistent with outcomes derived from the Indian Pines dataset, the classification accuracy of manifold learning algorithms based on the Pavia University dataset adheres to the same pattern of variation with respect to neighborhood size k and intrinsic dimensionality d, with the LTSA algorithm consistently outperforming other manifold learning methods in terms of classification accuracy.

The confusion matrices of each algorithm, as shown in Figure 19, clearly demonstrate that the experimental results are consistent with those obtained from the Indian Pines dataset. Validation with two datasets confirmed that the classification accuracy of the local tangent space alignment (LTSA) algorithm surpasses that of other manifold learning methods. This also verifies that the research findings of this paper are entirely correct.

2.Comparison of neighborhood computation time in nonlinear manifold learning

As revealed in Figure 20, compared to the Indian Pines dataset, the pattern of time consumption for neighborhood calculations by various manifold learning methods with changes in neighborhood size k and intrinsic dimensionality remains fundamentally consistent. However, a distinct observation is that the neighborhood computation time for the MVU algorithm on the Pavia University dataset shows minimal variation with an increase in the k value. This indicates that the impact of different hyperspectral image datasets on the feature extraction process using the MVU algorithm is significant.

3.Comparison of algorithm runtime in nonlinear manifold learning

As demonstrated in Figure 21, the runtime of various manifold learning methods based on the Pavia University dataset, when compared with the Indian Pines dataset, follows a fundamentally consistent pattern with changes in neighborhood size k and intrinsic dimensionality d.

## 4. Discussion

This study primarily investigates the performance of various manifold learning techniques in the dimensionality reduction of hyperspectral images and examines the patterns of how changes in the neighborhood size and intrinsic dimensionality affect the outcomes of dimensionality reduction. Below, we discuss and analyze the experimental results obtained in this study.

### 4.1. Discussion on the Visualization of Hyperspectral Image Dimensionality Reduction Using Manifold Learning Methods

Through the comparative visualization of dimensionality reduction by various manifold learning methods, the linear discriminant analysis (LDA) algorithm demonstrates superior intra-class and inter-class separability. In contrast, the principal component analysis (PCA) and multidimensional scaling (MDS) algorithms exhibit poorer inter-class separability, with clear separation between the edge regions of corn and forest, but almost all other categories overlap due to these algorithms’ insufficient consideration of local features. The Isomap algorithm shows better separability; however, the separability of other categories is poorer, which is determined by the algorithm’s acquisition of global coordinates. The locally linear embedding (LLE) algorithm effectively reflects both intra-class and inter-class separability. The separability of the Laplacian eigenmaps (LE) algorithm is poor, indicating that the construction of distance weights using the Gaussian kernel function significantly affects the classification results. The local tangent space alignment (LTSA) and Hessian locally linear embedding (HLLE) algorithms exhibit certain degrees of inter-class separability and poorer intra-class separability but clearly demonstrate the nonlinear relationships between high-dimensional features, determined by the algorithms’ maintenance of local properties and curvature. The maximum variance unfolding (MVU) algorithm’s results are similar to those of PCA and MDS. It can be concluded that when exploring the intrinsic nonlinear relationships of spectral features in hyperspectral images, nonlinear manifold learning methods outperform linear ones, with LTSA, LLE, and HLLE showcasing a better representation of the nonlinear relationships among spectral features of hyperspectral images.

### 4.2. Impact of Different Parameters on the Results of Feature Extraction from Hyperspectral Images Using Manifold Learning

Through comparative analysis of the feature extraction results from hyperspectral images using various manifold learning methods, this article illustrates that different parameters significantly impact the outcomes.

Since linear manifold learning methods do not incorporate a neighborhood construction phase, the classification outcomes solely depend on the determination of intrinsic dimensionality. By setting different intrinsic dimensions, it was found that the classification performance of LDA surpasses that of MDS and PCA. This superiority is attributed to LDA’s requirement for labeled sample information during the dimensionality reduction process, hence its enhanced performance, although it struggles with data possessing nonlinear structures. In the context of dimensionality reduction of hyperspectral datasets using nonlinear manifold learning, optimal neighborhood sizes k and intrinsic dimensions d exist, yet the optimal parameters differ across methods, dictated by whether the nonlinear manifold learning method bases its dimensionality reduction on global or local relational principles. Specifically for the LE and MVU algorithms, due to their unique characteristics, the outcomes of dimensionality reduction are significantly influenced by parameters t and λ. Therefore, in practical research, it is crucial to set appropriate values for t and λ based on research needs to either preserve global or local relationships. Comparative analysis of classification outcomes indicates that the LTSA algorithm outperforms other manifold learning methods, making it a preferred choice for studies on feature extraction from hyperspectral images.

### 4.3. The Impact of Intrinsic Dimension d and Neighborhood k on Neighborhood Computation and Overall Algorithm Runtime

Based on the analysis of neighborhood computation time and overall algorithm runtime as functions of neighborhood size and intrinsic dimensionality, it is observed that the Isomap and LE methods have a lower neighborhood computation time impact on the total runtime of the algorithm compared to the LLE method. Additionally, the dimension d has a minor effect on the neighborhood computation and runtime expenses for the LE, HLLE, and LTSA methods. However, the HLLE method incurs a significant total runtime cost due to the construction of quadratic terms required for dimensionality reduction, resulting from its higher computational complexity. The neighborhood computation time and overall runtime cost for the LTSA method are comparable, indicating a minor impact of neighborhood computation on the total runtime of the LTSA algorithm. The neighborhood computation time cost for the MVU method varies with k and d and is dependent on the dataset used; different datasets show significant variations in the neighborhood computation time for the MVU algorithm. Therefore, our study on the neighborhood computation time and overall runtime of various manifold learning algorithms will provide a reference and basis for future researchers investigating related issues.

### 4.4. Discussion on Other Widely Researched Hyperspectral Image Feature Extraction Methods

Currently, several other methods for feature extraction from hyperspectral imagery are widely used: feature extraction based on spatial domain algorithms, deep-learning-based feature extraction, band combination-based feature extraction, and index parameter-based feature extraction. However, feature extraction methods based on spatial domain algorithms suffer from large computational requirements for statistical models and difficulties in selecting model parameters. Deep-learning-based feature extraction possesses strong fitting capabilities, but with the complexity of network structures, it can lead to overfitting and is highly susceptible to falling into local minima. Band combination-based feature extraction can only group consecutive bands as one, limiting the improvement of its discriminative performance. Index parameter-based feature extraction requires domain-specific expert knowledge. Manifold learning methods can effectively express low-dimensional manifold structures embedded in high-dimensional spaces, but these methods are significantly influenced by parameters. Thus, this forms the purpose and significance of our study.

### 4.5. Uncertainties, Limitations, and Future Direction

#### 4.5.1. Uncertainties

(1) Model assumption uncertainty: Due to the influence of equipment and external environmental factors on hyperspectral data acquisition, the distribution of data points may not be entirely situated within a single manifold space; instead, there may exist two or more manifold spaces. Consequently, the fundamental assumptions of the model might introduce uncertainty, affecting the effectiveness of dimensionality reduction.

(2) Uncertainty in parameter selection:

Choice of neighborhood k: The size of k directly impacts the representation of the manifold’s local structure. A k value that is too small might fail to capture sufficient local information, whereas k value that is too large could introduce noise or irrelevant features, leading to poorer dimensionality reduction outcomes. Different datasets and manifold learning algorithms have varying optimal values for k, lacking a unified standard for selection.

Selection of intrinsic dimension d: The choice of intrinsic dimension relates to the richness of data representation in a low-dimensional space. Setting d too low could result in the loss of crucial information; conversely, setting d too high might introduce noise or meaningless dimensions.

(3) Uncertainty of the algorithms themselves: Different manifold learning algorithms may produce varying dimensionality reduction outcomes when processing the same data. This uncertainty stems from the internal mathematical mechanisms and optimization strategies of the algorithms. For instance, LLE focuses on preserving the distances between local neighboring points, whereas Isomap aims to maintain global geodesic distances.

(4) Data uncertainty: The complexity and diversity of hyperspectral data also introduce uncertainty into manifold learning. For instance, the spectral similarity between different ground objects, noise and outliers in the data, as well as the unevenness of sampling density, can all affect the quality of dimensionality reduction and the optimization of parameter selection.

(5) Uncertainty in computational complexity and scalability: As the volume of data increases, the computational complexity of manifold learning algorithms also significantly rises. For large-scale hyperspectral datasets, the scalability and efficiency of algorithms become crucial considerations, which may limit the application scope of some computationally intensive algorithms.

#### 4.5.2. Limitations

(1) Research scope of manifold learning methods: Although this study has attempted to include as many existing manifold learning methods as possible, it has not encompassed all available methods due to certain reasons. Therefore, a comprehensive study and comparative analysis of all manifold learning methods for hyperspectral feature extraction, such as t-SNE and UMAP, have not been conducted.

(2) Limitations of model assumptions: The manifold learning methods are based on the assumption that hyperspectral data are distributed on a low-dimensional manifold. Although this assumption is reasonable in many cases, it may not apply universally to all hyperspectral data. The intrinsic structure of complex data might be more complicated than a low-dimensional manifold, potentially limiting the applicability and performance of the model.

(3) Difficulty in parameter selection: In this paper, parameters are selected primarily based on existing research literature, through cross-validation, or by repeated adjustment according to the classification results in the experimental phase until final determination, rather than achieving adaptive parameter selection.

(4) Computational complexity: Though some manifold learning methods have shown excellent performance on small datasets, their computational complexity becomes significantly high when applied to large-scale hyperspectral data, demanding extensive computational resources. This limits their practical applicability on a large scale.

(5) Robustness issues: Manifold learning methods are sensitive to noise and outliers, which can degrade the quality of dimensionality reduction. Especially in hyperspectral imagery, the uncertainty in data quality due to sensor noise, atmospheric interference, etc. may increase instability in the dimensionality reduction process.

(6) Limitations in cross-domain applications: Current research and applications of manifold learning methods are mainly focused on specific data types or domains. These methods may require targeted adjustments or optimizations for different types of data or cross-domain applications, limiting their widespread use.

#### 4.5.3. Future Direction

The experimental results have demonstrated the effectiveness of the algorithms. Although manifold learning-based feature extraction applications for hyperspectral imaging have achieved substantial results in the past few years, the complexity of the mathematical theoretical foundation, along with the influence of acquiring hyperspectral images from multiple platforms and sensors under highly complex external environmental conditions and the intersection and integration across related disciplines, remain issues worthy of further exploration.

(1) Development of improved manifold learning algorithms: In response to the uncertainties of existing algorithms, such as uncertainties in model assumptions and parameter selection, future research can focus on developing new or improved manifold learning algorithms. These algorithms should more accurately capture the intrinsic structure of hyperspectral data while providing more flexible and adaptive parameter setting mechanisms.

(2) Big data and computational efficiency: As the volume of hyperspectral data increases, future manifold learning algorithms need to pay close attention to computational efficiency and scalability. The development of distributed computing and parallel processing methods and the utilization of modern hardware architectures (such as GPU acceleration) to handle large-scale hyperspectral datasets are essential.

(3) Robustness and noise handling: Given the complexity of hyperspectral data, new manifold learning algorithms should improve robustness to noise and outliers. Research on effectively identifying and handling these data can enhance the accuracy and reliability of dimensionality reduction results.

(4) Integration of manifold learning algorithms with various technologies and fusion with multi-source data: It is noteworthy that in the past two years, with the introduction and successful application of more novel and effective machine-learning and pattern analysis algorithms, several manifold learning methods that are more closely integrated with machine-learning techniques have been published. Some examples include neighborhood feature preservation methods based on local discriminant analysis [44], multi-feature manifold learning methods [45], and tensor manifold learning methods in high-order feature spaces [46], among others. Future research could leverage GeoAI technology, combining machine learning and deep learning to enhance the efficiency and accuracy of feature extraction in manifold learning models. Effectively integrating the strengths of deep learning, with its powerful learning capabilities and efficient feature representation, could further improve algorithm performance and foster the development of novel manifold learning algorithms. Additionally, future studies might consider fusing hyperspectral data with other multi-source remote-sensing data to enrich information acquisition, potentially augmenting the performance of nonlinear manifold learning algorithms in tasks such as land cover classification and object recognition.

## 5. Conclusions

(1) This article is grounded in the theory of differential manifolds and the relevant definitions of differential geometry, providing a formalized definition of dimensionality reduction and feature extraction for hyperspectral images, as well as the process of feature extraction through manifold learning. By employing manifold learning for the three-dimensional visualization of hyperspectral images, it has been determined that nonlinear manifold learning methods possess certain advantages in capturing the intrinsic nonlinear structural relationships within hyperspectral imagery. The classification results indicate that the parameters, neighborhood k and intrinsic dimension d, have a significant impact on feature extraction using nonlinear manifold learning methods, with the LTSA algorithm exhibiting higher classification accuracy. This section’s research contributes valuable insights into the principles of dimensionality reduction for hyperspectral data through manifold learning and offers significant guidance for the selection of manifold learning algorithms in hyperspectral feature extraction.

(2) Comparing the neighborhood computation time and total runtime for feature extraction and dimensionality reduction using different manifold learning methods in hyperspectral imaging, it was found that the HLLE algorithm has both a neighborhood computation time and a total runtime that are greater than those of other manifold methods, indicating a higher computational complexity for HLLE. In contrast, the LE algorithm’s neighborhood computation time and total runtime are significantly lower than those of other manifold methods, suggesting that LE has a very low computational complexity. The computational complexities of other nonlinear manifold learning methods fall between these two algorithms. This section’s findings are of significant reference value when considering the time expenditure for dimensionality reduction.

(3) Under the influence of local structure weight λ and Gaussian kernel bandwidth t, analysis of the MVU and LE algorithms demonstrates that smaller t values emphasize the local structure of the data, whereas smaller λ values highlight the global structure. Conversely, larger t values give precedence to global structures, whereas larger λ values emphasize local structures. Specifically, the MVU algorithm’s classification results after dimensionality reduction are greatly influenced by λ and neighborhood size k when facing different datasets. This section provides theoretical guidance for studying high-dimensional data dimensionality reduction using the MVU and LE algorithms.

## Figures and Tables

**Figure 1 sensors-24-02089-f001:**
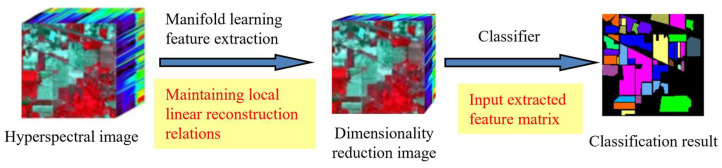
Dimensionality reduction and classification process of hyperspectral images based on manifold learning.

**Figure 2 sensors-24-02089-f002:**
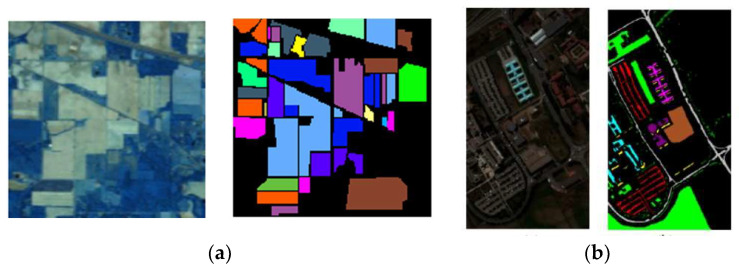
False color image and ground truth: (**a**) Indian Pines data; (**b**) Pavia University data.

**Figure 3 sensors-24-02089-f003:**
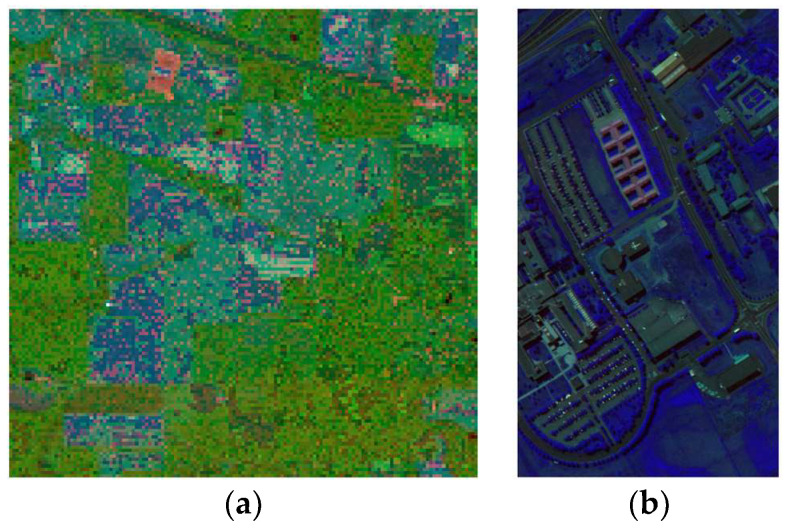
RGB pseudocolor image: (**a**) Indian Pines data; (**b**) Pavia University data. The dataset comprises a total of 21,025 pixels, of which only 10,249 pixels are land cover pixels, and the remaining 10,776 pixels are background pixels. In practical classification tasks, these background pixels need to be excluded. The distribution of land cover pixels is presented in Table 1.

**Figure 4 sensors-24-02089-f004:**
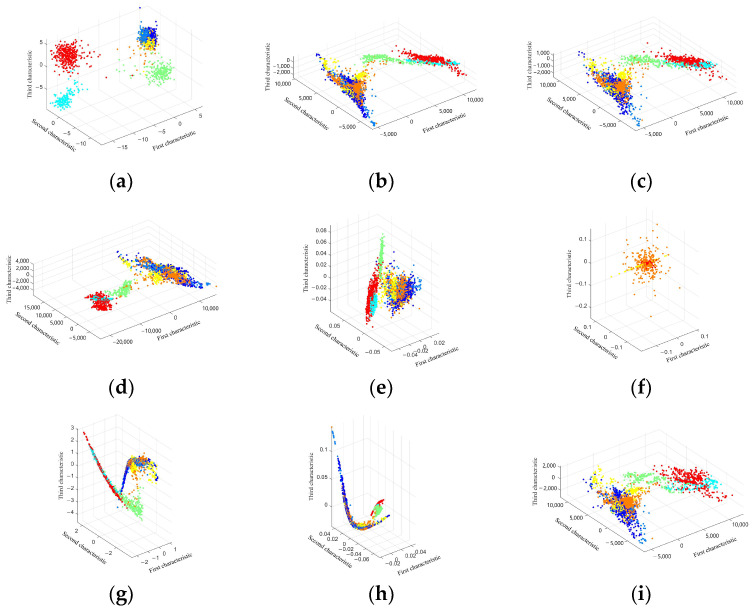
Low-dimensional embedding performance of Indian Pines dataset using different manifold learning methods: (**a**) LDA; (**b**) MDS; (**c**) PCA; (**d**) Isomap; (**e**) LLE; (**f**) LE; (**g**) HLLE; (**h**) LTSA; (**i**) MVU.

**Figure 5 sensors-24-02089-f005:**
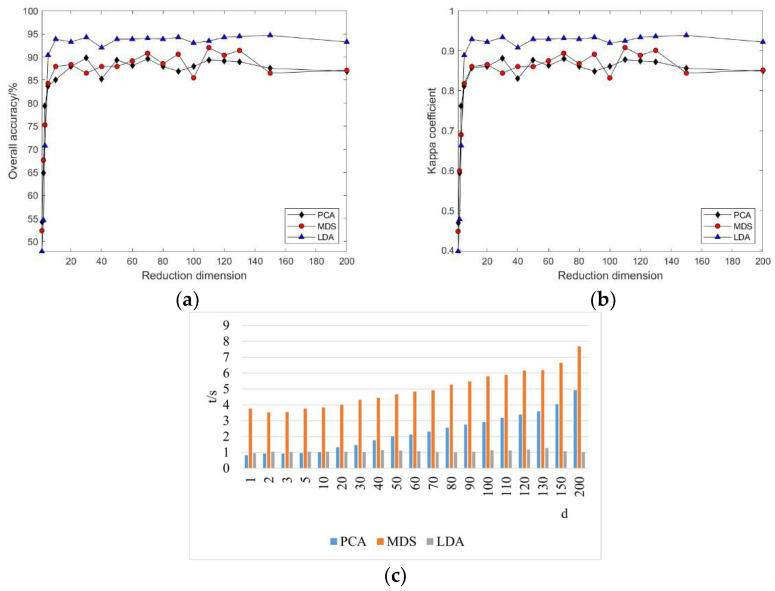
Comparison of dimensionality reduction of Indian Pines datasets using different manifold learning methods as the intrinsic dimension d increases: (**a**) overall accuracy; (**b**) Kappa coefficient; (**c**) algorithm runtime.

**Figure 6 sensors-24-02089-f006:**
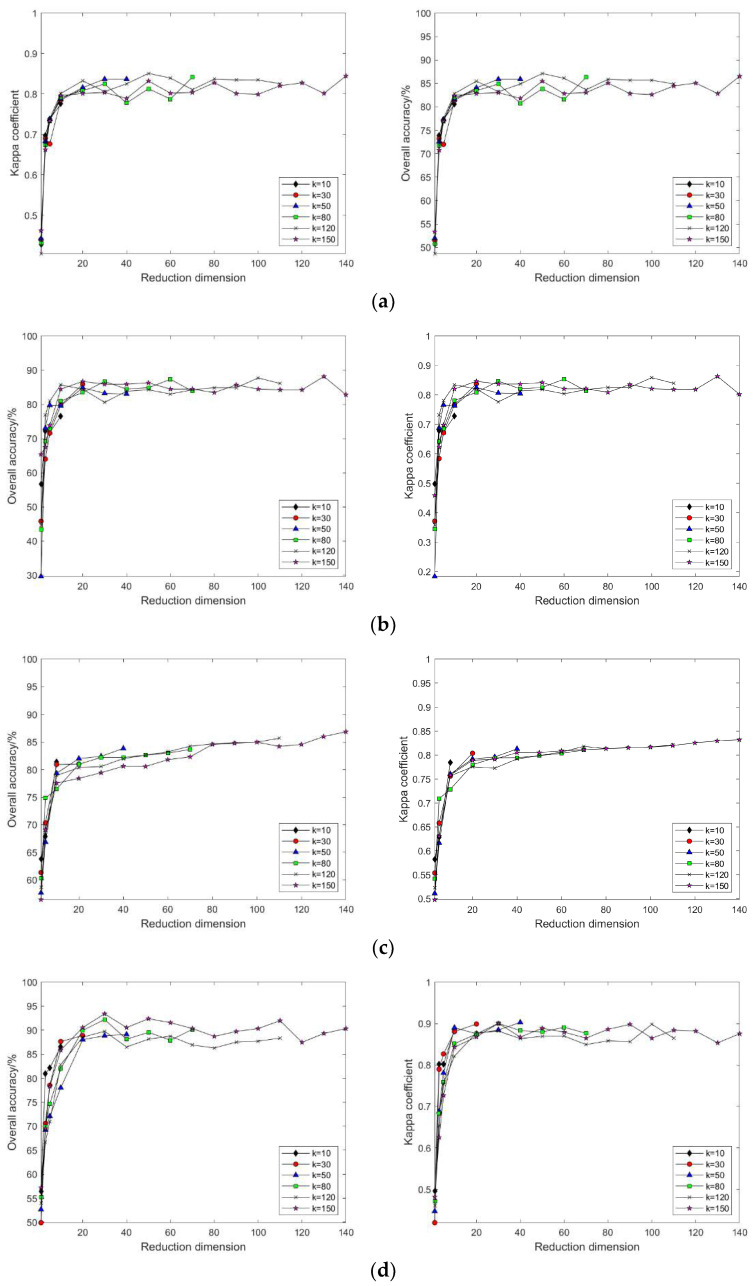
Variation in overall accuracy and Kappa coefficient with increasing intrinsic dimension d when different manifold learning methods take different neighborhood k values: (**a**) Isomap; (**b**) LLE; (**c**) HLLE; (**d**) LTSA.

**Figure 7 sensors-24-02089-f007:**
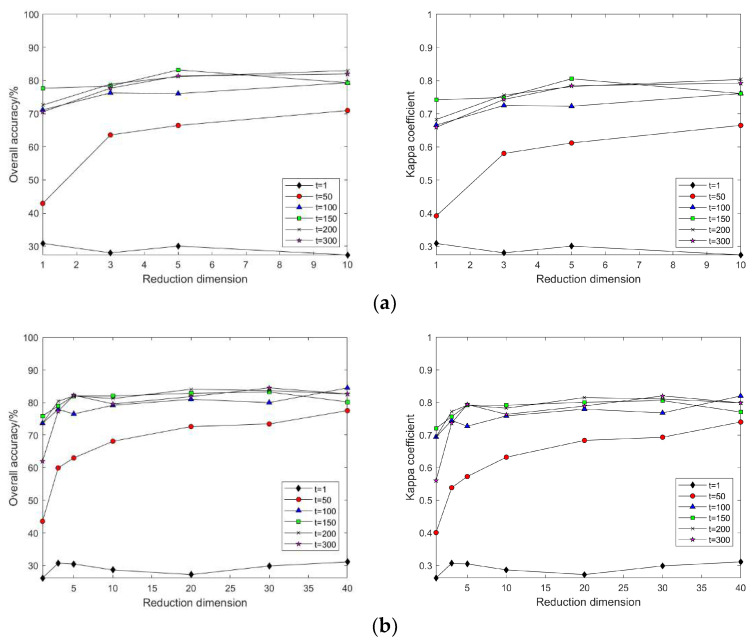
Variation in LE algorithm’s land feature classification accuracy and Kappa coefficient with different values of k and increasing intrinsic dimension d as t takes different values: (**a**) k = 10; (**b**) k = 50; (**c**) k = 80; (**d**) k = 120; (**e**) k = 150.

**Figure 8 sensors-24-02089-f008:**
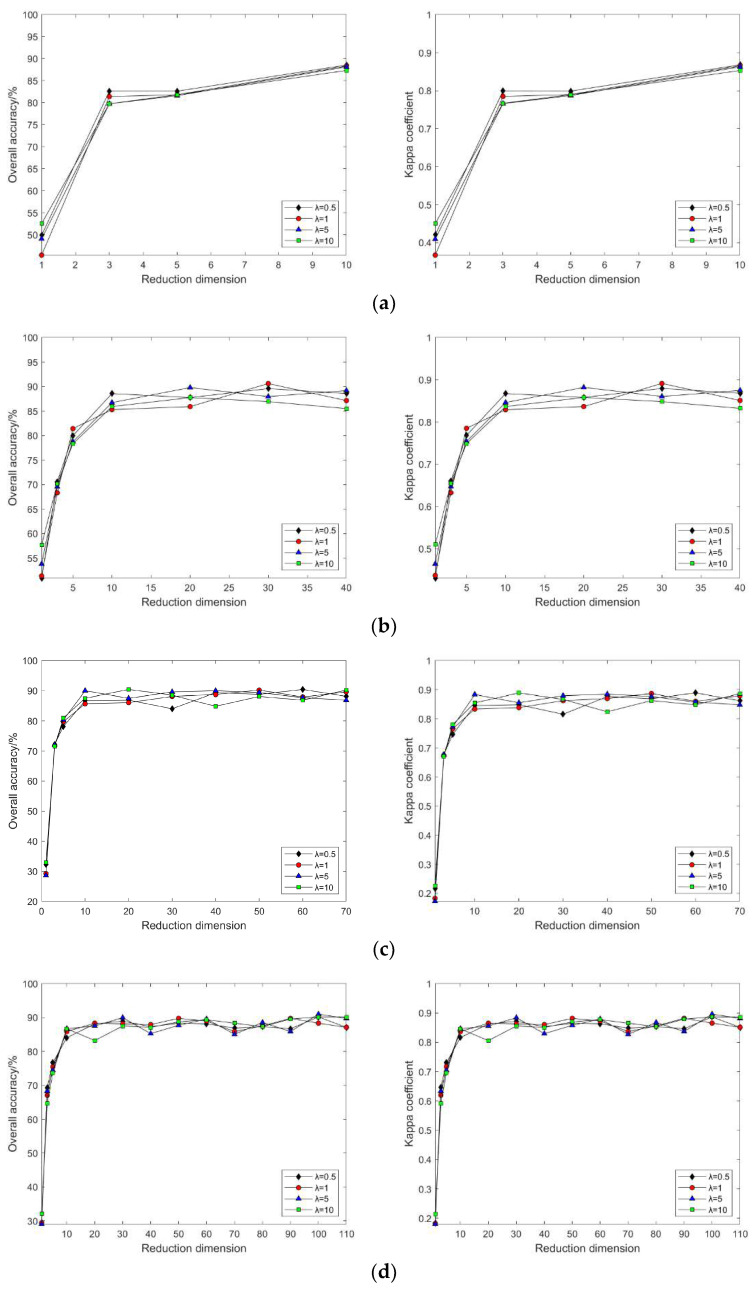
When taking different k values and increasing them with the intrinsic dimension d, the accuracy of the MVU algorithm in land use classification and the Kappa coefficient are different λ. The variation pattern of values (**a**) k = 10; (**b**) k = 50; (**c**) k = 80; (**d**) k = 120; (**e**) k = 150.

**Figure 9 sensors-24-02089-f009:**
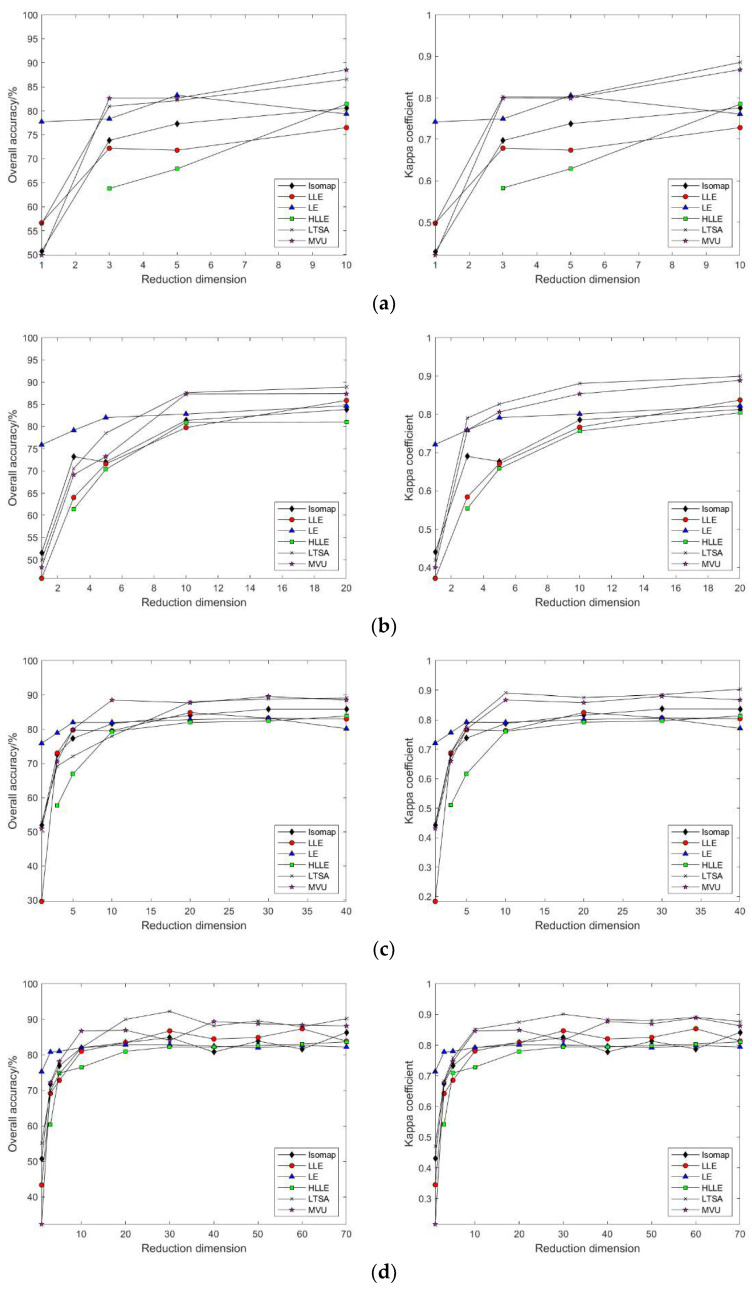
Take different k values and compare the overall accuracy and Kappa coefficients of various manifold learning methods as the intrinsic dimension d increases: (**a**) k = 10; (**b**) k = 30; (**c**) k = 50; (**d**) k = 80; (**e**) k = 120; (**f**) k = 150.

**Figure 10 sensors-24-02089-f010:**
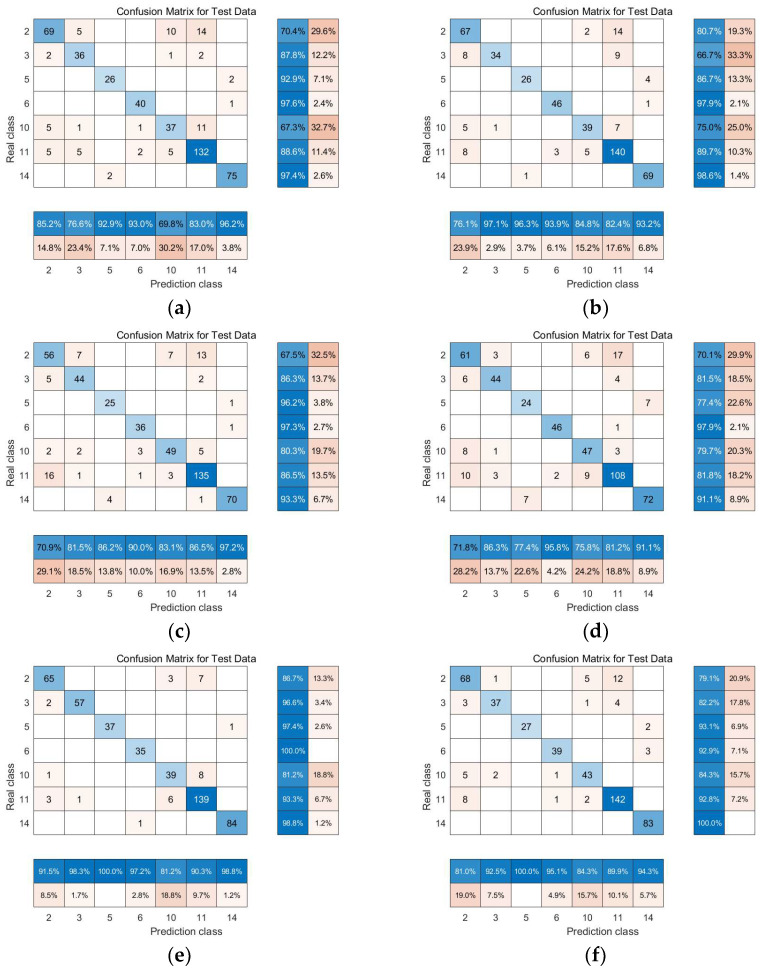
Confusion matrices for land cover classification using various manifold learning methods: (**a**) Isomap; (**b**) LLE; (**c**) LE; (**d**) HLLE; (**e**) LTSA; (**f**) MVU.

**Figure 11 sensors-24-02089-f011:**
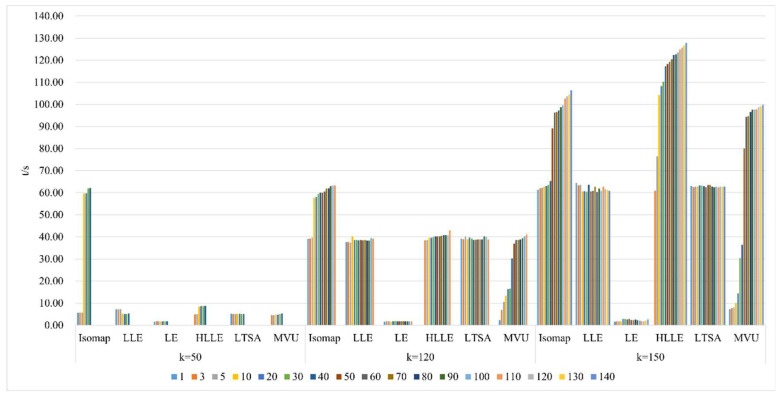
Based on the Indian Pines dataset, take a certain value of k and compare the computation time of the best neighborhood of manifold learning algorithms as the dimension d increases..

**Figure 12 sensors-24-02089-f012:**
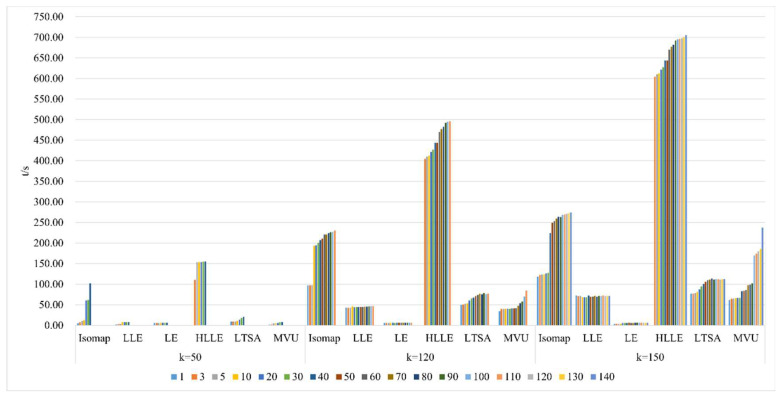
Based on the Indian Pines dataset, take a certain value of k and compare the running time of manifold learning algorithms as the dimension d increases.

**Figure 13 sensors-24-02089-f013:**
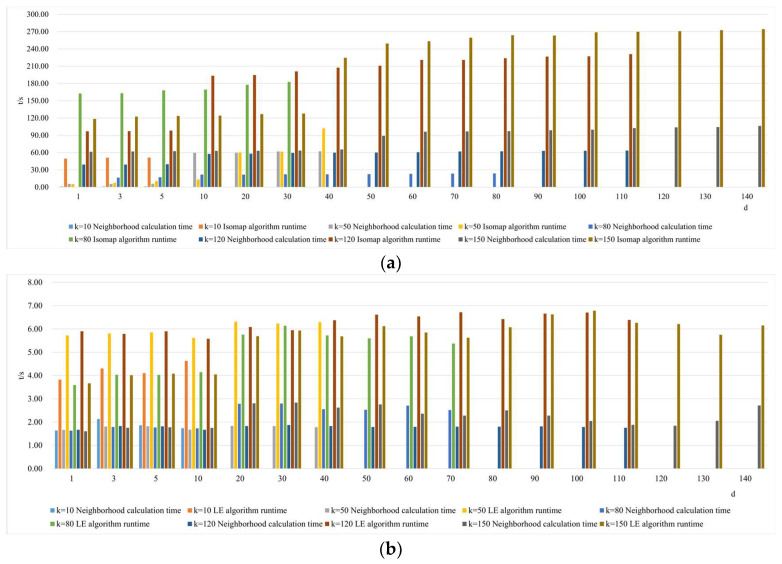
Taking a certain value of k, comparing the optimal neighborhood and algorithm running time of different manifold learning methods as d increases: (**a**) Isomap; (**b**) LE; (**c**) LLE; (**d**) HLLE; (**e**) LTSA; (**f**) MVU.

**Figure 14 sensors-24-02089-f014:**
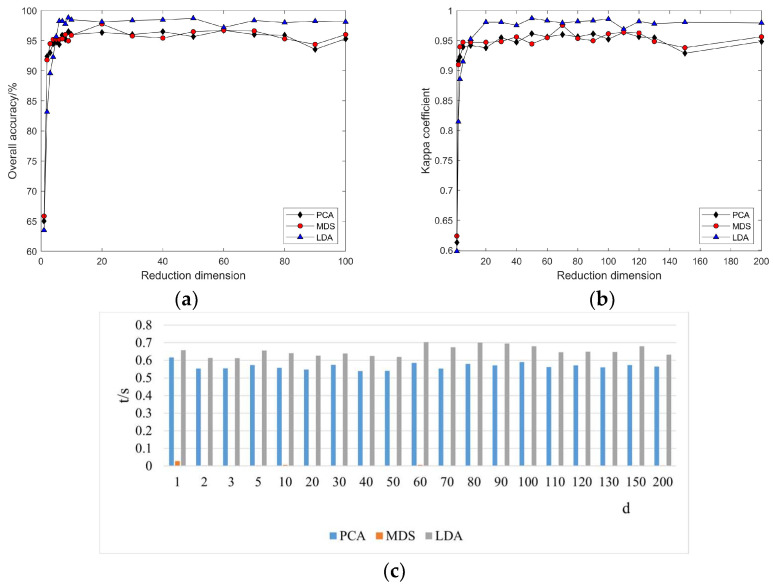
Comparison of dimensionality reduction of Pavia University datasets using different manifold learning methods as the intrinsic dimension d increases: (**a**) overall accuracy; (**b**) Kappa coefficient; (**c**) algorithm runtime.

**Figure 15 sensors-24-02089-f015:**
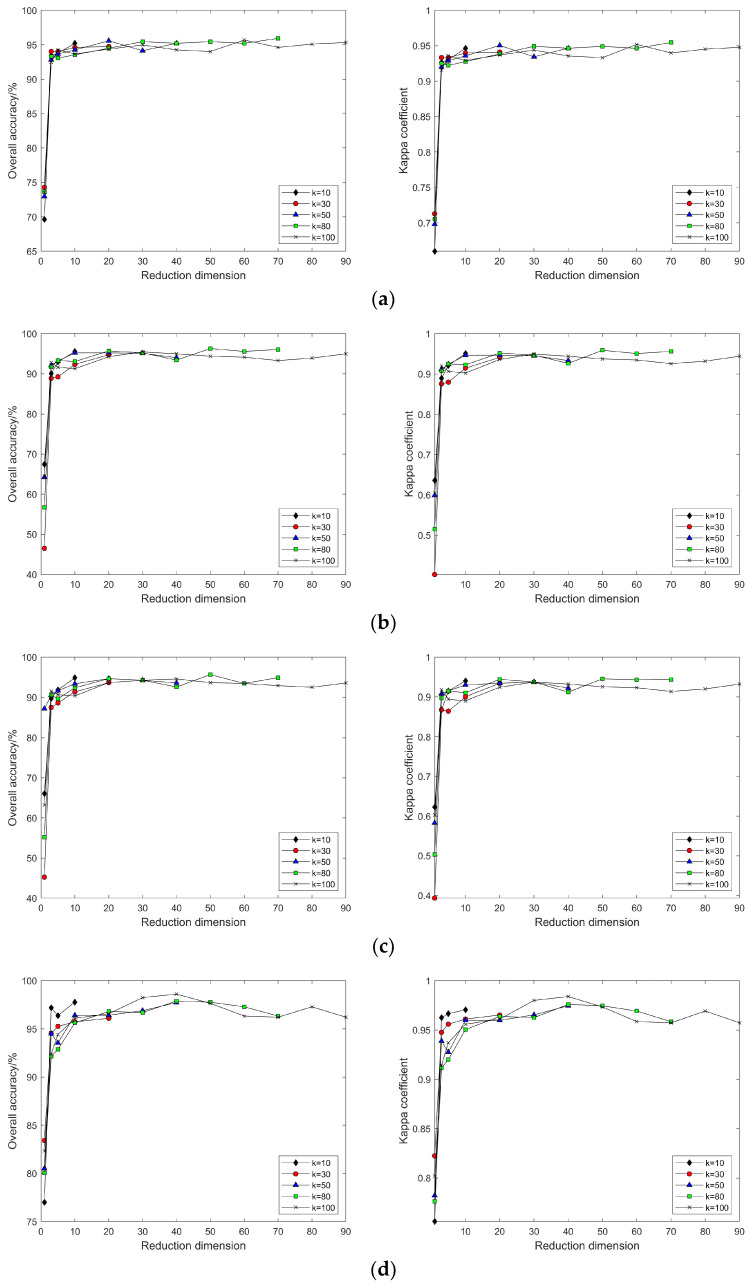
Variation in overall accuracy and Kappa coefficient with increasing intrinsic dimension d when different manifold learning methods take different neighborhood k values: (**a**) Isomap; (**b**) LLE; (**c**) HLLE; (**d**) LTSA.

**Figure 16 sensors-24-02089-f016:**
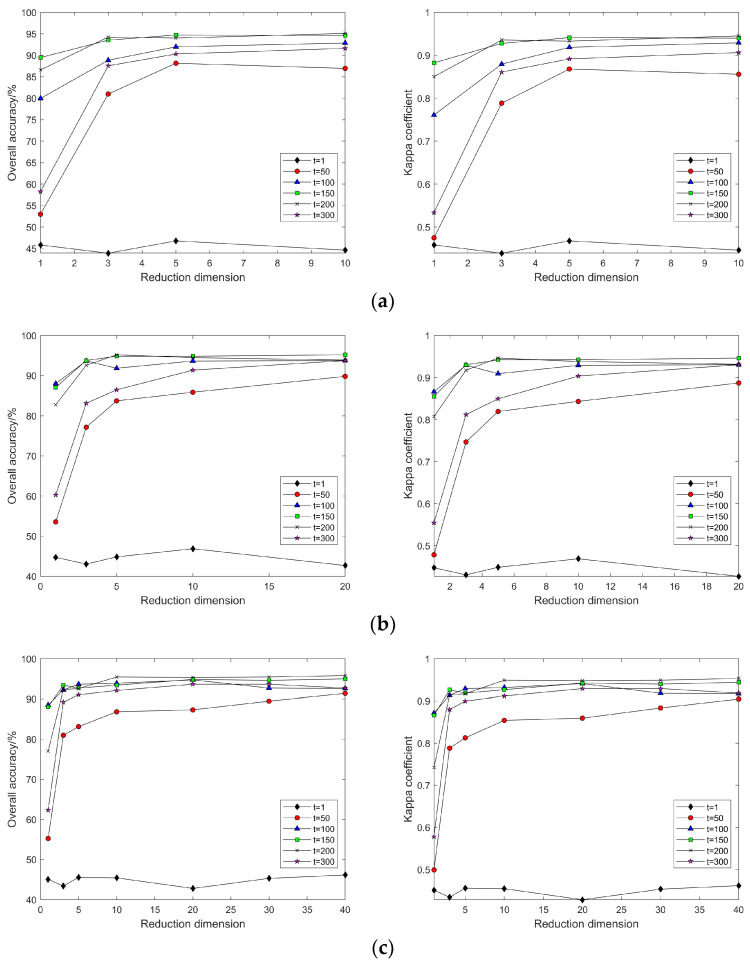
Variation in the LE algorithm’s land feature classification accuracy and Kappa coefficient with different values of k and increasing intrinsic dimension d as t takes different values: (**a**) k = 10; (**b**) k = 30; (**c**) k = 50; (**d**) k = 80; (**e**) k = 100.

**Figure 17 sensors-24-02089-f017:**
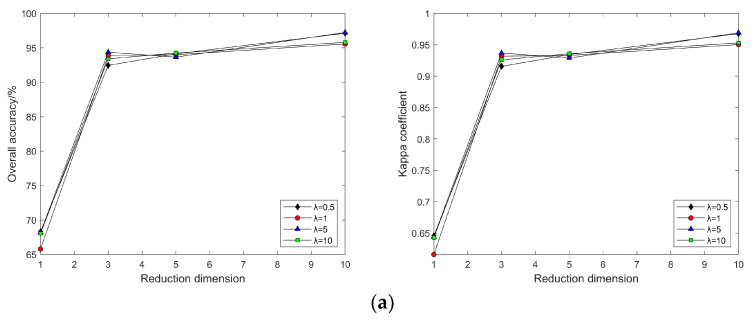
When taking different k values and increasing them with the intrinsic dimension d, the accuracy of the MVU algorithm in land use classification and the Kappa coefficient are different λ. The variation pattern of values: (**a**) k = 10; (**b**) k = 30; (**c**) k = 50; (**d**) k = 80; (**e**) k = 100.

**Figure 18 sensors-24-02089-f018:**
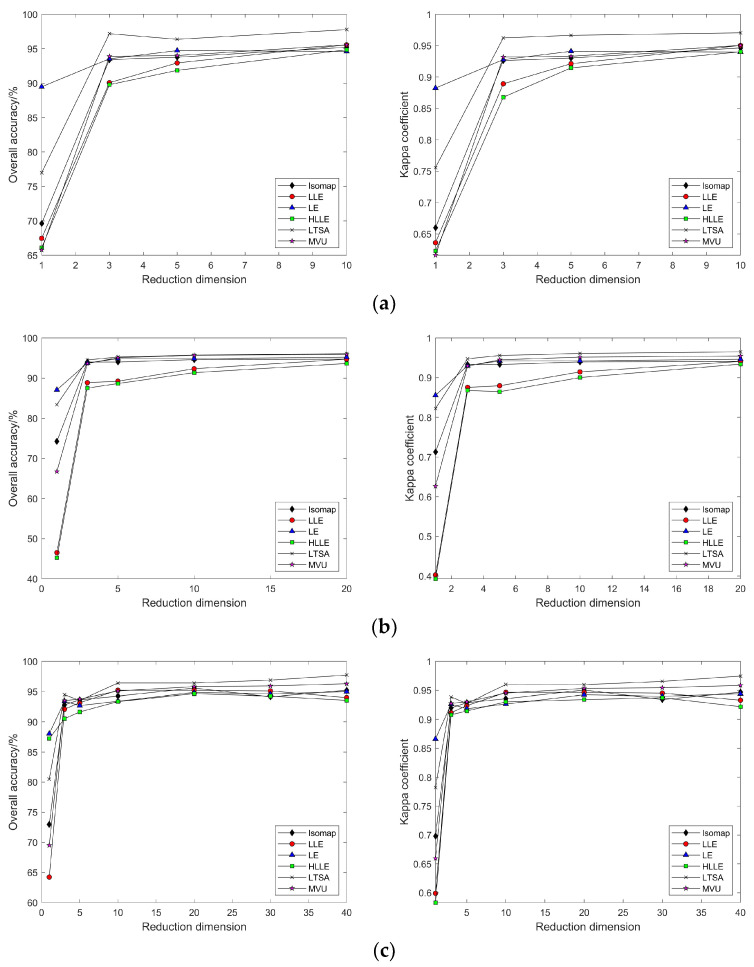
Take different k values and compare the overall accuracy and Kappa coefficients of various manifold learning methods as the intrinsic dimension d increases: (**a**) k = 10; (**b**) k = 30; (**c**) k = 50; (**d**) k = 80; (**e**) k = 100.

**Figure 19 sensors-24-02089-f019:**
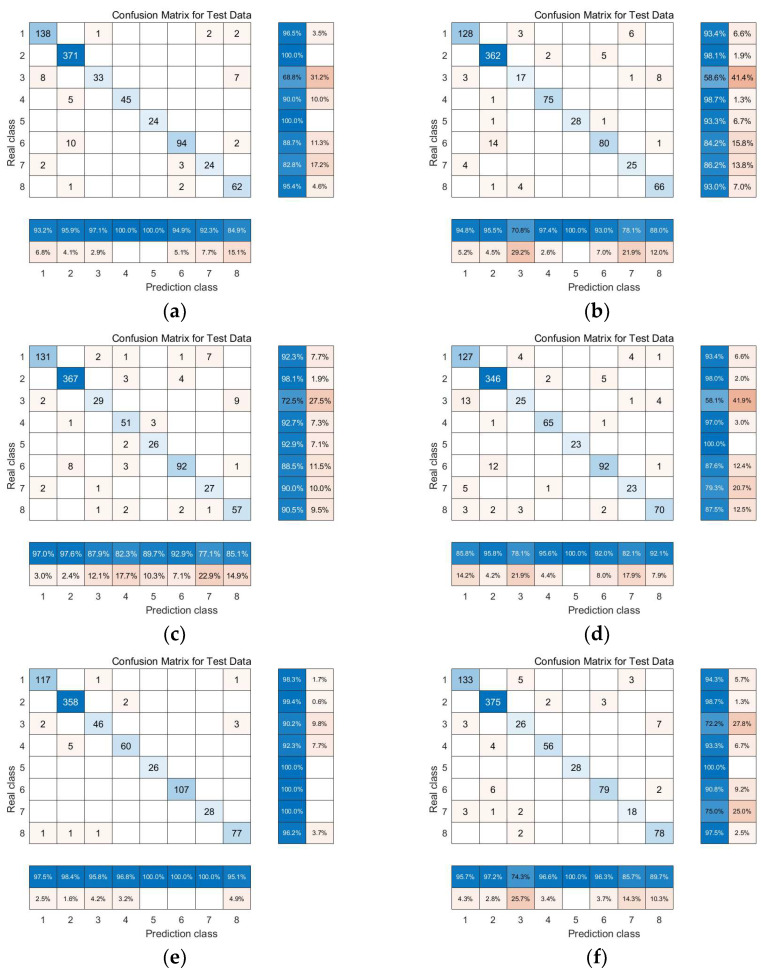
Confusion matrices for land cover classification using various manifold learning methods: (**a**) Isomap; (**b**) LLE; (**c**) LE; (**d**) HLLE; (**e**) LTSA; (**f**) MVU.

**Figure 20 sensors-24-02089-f020:**
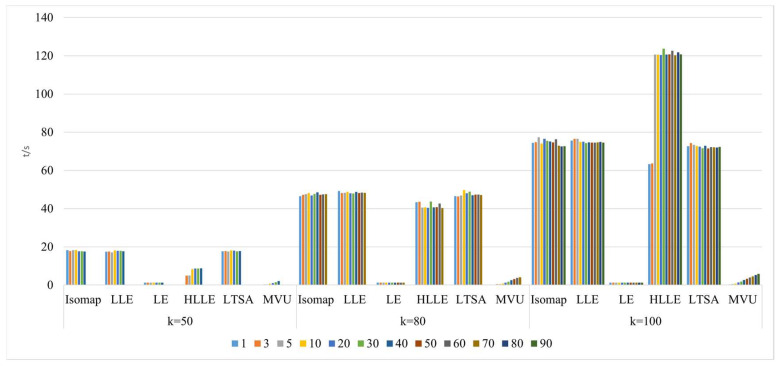
Based on the Pavia University dataset, take a certain value of k and compare the computation time of the best neighborhood of manifold learning algorithms as the dimension d increases.

**Figure 21 sensors-24-02089-f021:**
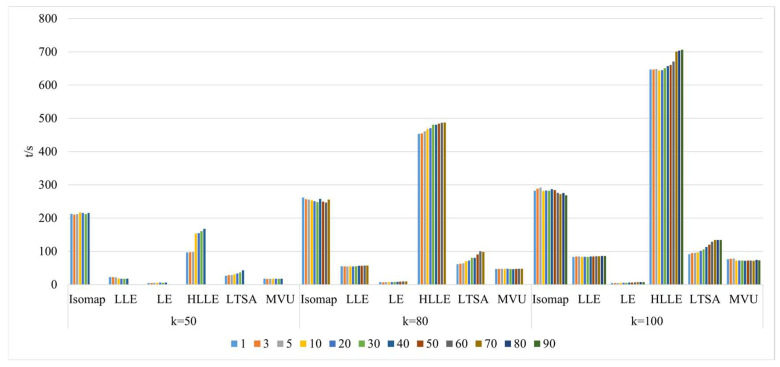
Based on the Pavia University dataset, take a certain value of k and compare the running time of manifold learning algorithms as the dimension d increases.

**Table 1 sensors-24-02089-t001:** Ground truth classes and sample numbers of Indian Pines.

Category	Classification	Color	Samples
1	Alfalfa		46
2	Corn-notill		1428
3	Corn-mintill		830
4	Corn		237
5	Grass-pasture		483
6	Grass-tree		730
7	Grass-pasture-mowed		28
8	Hay-windrowed		478
9	Oats		20
10	Soybean-notill		972
11	Soybean-mintill		2455
12	Soybean-clean		593
13	Wheat		205
14	Woods		1265
15	Buildings-Grass-Tress-Drives		386
16	Stone-Steel-Towers		93

**Table 2 sensors-24-02089-t002:** Ground truth classes and sample numbers of Pavia University.

Category	Classification	Color	Samples
1	Asphalt		6631
2	Meadows		18,649
3	Gravel		2099
4	Trees		3064
5	Painted metal sheets		1345
6	Bare Soil		5029
7	Bitumen		1330
8	Self-Blocking Bricks		3682
9	Stone-Steel-Towers		947

**Table 3 sensors-24-02089-t003:** The total number of samples for both datasets and the division of data samples selected for study in this paper.

Indian Pines Dataset				Pavia University Dataset			
Category	Classification	Total Number of Samples	Number of Samples Selected	Category	Classification	Total Number of Samples	Number of Samples Selected
2	Corn-notill	1428	428	1	Asphalt	6631	663
3	Corn-mintill	830	249	2	Meadows	18,649	1865
5	Grass-pasture	483	145	3	Gravel	2099	210
6	Grass-tree	730	219	4	Trees	3064	306
10	Soybean-notill	972	292	5	Painted metal sheets	1345	135
11	Soybean-mintill	2455	737	6	Bare Soil	5029	503
14	Woods	1265	380	7	Bitumen	1330	133
——	——	——	——	8	Self-Blocking Bricks	3682	368
Total	——	8163	2449	Total	——	41,829	4183

**Table 4 sensors-24-02089-t004:** Division of training and validation sets for classification experiments using the Indian Pines and Pavia University datasets.

Indian Pines Dataset				Pavia University Dataset			
Category	Classification	Training Set	Validation Set	Category	Classification	Training Set	Validation Set
2	Corn-notill	342	86	1	Asphalt	530	133
3	Corn-mintill	199	50	2	Meadows	1492	373
5	Grass-pasture	116	29	3	Gravel	168	42
6	Grass-tree	175	44	4	Trees	245	61
10	Soybean-notill	234	58	5	Painted metal sheets	108	27
11	Soybean-mintill	590	147	6	Bare Soil	402	101
14	Woods	304	76	7	Bitumen	106	27
——	——	——	——	8	Self-Blocking Bricks	295	74
Total	——	1959	490	Total	——	3346	837

## Data Availability

The hyperspectral remote-sensing images from the Indian Pines dataset were obtained from https://www.ehu.eus/ccwintco/index.php/Hyperspectral_Remote_Sensing_Scenes (accessed on 9 January 2024).

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
