# Peer review of "A Study on Dimensionality Reduction and Parameters for Hyperspectral Imagery Based on Manifold Learning"

_sensors, 2024, doi:10.3390/s24072089_

Round 1

Reviewer 1 Report

Comments and Suggestions for Authors

The presented work is an actual and perspective study. Indeed, with the rapid development of remote sensing and information technologies, hyperspectral imaging has become widely used by researchers. In this connection, the problem naturally arises of reducing the dimensionality of hyperspectral data in order to increase the efficiency of analysis. The authors examined the possibilities of feature extraction for hyperspectral data using a variety of approaches. The advantage of the work is the analysis of the optimal time for calculating neighborhoods and the total execution time of the algorithms.

There are several recommendations for work:

1. It is recommended to clearly formulate the gap in the introduction and objectives.

2. It would be better to present the algorithms of the methods in the classical style, step by step "1, 2, ..."

Reviewer 2 Report

Comments and Suggestions for Authors

The manuscript describes an investigation of different manifold learning methods to reduce data dimensionality in hyperspectral imaging data. While the theme complex is well known to experts in the field the manuscript provides an interesting overview over the different methods and their merit when applied to hyperspectral data.

The manuscript is well written and presents the data well. However, the structure of the manuscript needs to be significantly improved to be considered for publication. Parts of the manuscript – especially discussion and conclusion – read like bullet points. There is also a high degree of repetition and redundance within the text, especially in the introduction and results. The first two sections of the results fail to present any results of the study and would be better placed in the introduction or methodology chapters. Introduction and results could overall be condensed. Large parts of the discussion summarizes the results of the experiments without truly discussing them and in the entire discussion is no reference to other scientific studies and their findings for dimensionality reduction in hyperspectral imaging.

Reviewer 3 Report

Comments and Suggestions for Authors

-Although the novelty of this topic is not high, it has certain significance for hyperspectral dimension reduction and classification, and the workload is larger. However, the paper lacks a precise description of the data, and some experimental steps and meanings are unclear.

Here are the detailed recommendations:

-Line 472 It is difficult to intuitively understand the dimensionality reduction process of hyperspectral image based on the image which is too roughï¼›

-This paper does not mention what classifier is used for sample classification, and the selection of classifier often has a certain impact on the classification results.

-It is suggested to add the visual chart of the original hyperspectral data, such as selecting the original three-band data with low correlation to form a visual chart to provide a reference for comparison;

-This paper lacks the method of dividing training set, test set and verification set in experimental data.

- Line643-652 The selection of T-value classification accuracy diagram is inconsistent with that in the kappa coefficient diagram.

-What is the basis for the selection of neighborhood k, and whether the value of k is reasonable;

-In this paper, seven ground objects were selected for experiment, but only the overall classification accuracy and kappa coefficient were found in the results. In addition, there is no direct view of the classification results, and it is impossible to see whether each pixel is eventually classified correctly.

Comments on the Quality of English Language

Minor editing of English language required

Reviewer 4 Report

Comments and Suggestions for Authors

The problems in the paper are summarized as follows:

1.Figure 2 is too basic to represent the schematic diagram of hyperspectral image dimensionality reduction based on  manifold learning technology. Suggest the author to make modifications to reflect the specific details and steps of the experiment.

2.The formula writing is not standardized, and the matrix with 285 lines needs to be bolded. Additionally, the formula needs to be labeled instead of written in the main text.

3.It is recommended to add at least one dataset of hyperspectral images to demonstrate the validity and universality of the conclusions.

Comments on the Quality of English Language

Minor editing of English language required

Round 2

Reviewer 2 Report

Comments and Suggestions for Authors

The authors significantly improved the quality of the manuscript and the reviewer would like to thank them for their efforts.

Reviewer 3 Report

Comments and Suggestions for Authors

The authors carefully responded to the questions and suggestions raised, corrected the errors in the paper, and added the necessary diagrams to show the details of the experimental process and results. The work of this paper is relatively large, and it has certain reference significance in the field of hyperspectral dimension reduction, so I agree to accept in present form.

Reviewer 4 Report

Comments and Suggestions for Authors

Accept in present form

Comments on the Quality of English Language

Minor editing of English language required